# Executable Agentic Memory for GUI Agent

Zerui Qin[1]  Sheng Yue[2]  Xingyuan Hua[1]  Yongjian Fu[1]  Ju Ren[1]

## Abstract

Modern GUI agents typically rely on a model-centric and step-wise interaction paradigm, where LLMs must re-interpret the UI and re-decide actions at every screen, which is fragile in long-horizon tasks. In this paper, we propose Executable Agentic Memory (EAM), a structured Knowledge Graph (KG) that shifts GUI planning from free-form generation to a robust retrieval-and-execution process. Our approach includes a sample-efficient memory construction pipeline using state-aware DFS and action-group mining to compress multi-step routines. To ensure efficient planning, we introduce a value-guided graph search where a lightweight Q-function model steers Monte Carlo Tree Search (MCTS) over the KG. We theoretically establish bias-consistency for the Q-model and derive sample complexity bounds for path recovery. Empirically, EAM outperforms state-of-the-art baselines like UI-TARS-7B by up to $19.6\%$ on AndroidWorld, while reducing token costs $6\times$ relative to GPT-4o. With a $2.8$s average latency, EAM enables reliable, quick, and long-horizon GUI automation. Our code is available at https://github.com/ZeruiQin/EAM.

## 1. Introduction

Modern Graphical User Interface (GUI) agents powered by (multimodal) LLMs can operate real-world apps by "seeing" screens and generating actions (Wen et al., 2024; Wang et al., 2024a; Zhang et al., 2025). However, the dominant interaction paradigm remains *model-centric and step-wise*: at every screen, an LLM must re-interpret the UI, re-decide the next action, and implicitly maintain task progress in its context window. This makes long-horizon automation frag-ile: small perceptual or reasoning errors would compound, easily producing hallucinated actions and incorrect detours, especially in heterogeneous app environments where training coverage is limited (Qin et al., 2025; Luo et al., 2025; Wu et al., 2025; Gou et al., 2024). For instance, UI-TARS-7B (Qin et al., 2025), considered a SOTA GUI agent model, achieves only 33% success rate on the long-horizon AndroidWorld benchmark (Rawles et al., 2024), while M3A, an agentic framework powered by GPT-4o, attains merely 40.5% (Rawles et al., 2024).

To improve robustness, a natural direction is to equip agents with external knowledge and memory. Some efforts (Wang et al., 2024b; 2025; Cheng et al., 2025; Sun et al., 2026) maintain textual memory of historical interactions, such as workflow patterns and decision heuristics, and inject them into the LLM's context to guide task planning. Others construct external knowledge bases by extracting action-level knowledge (e.g., element functionality or successful trajectories) from exploration, storing them as vector databases or knowledge graphs, and retrieving relevant knowledge at inference time to augment decision-making (Xie et al., 2025; Jiang et al., 2025; Guan et al., 2025b; Li et al., 2025). However, such in-context knowledge injection remains unreliable due to model-centric generation and ignorance of inherent structured information in historical trajectories, making it difficult to reliably reproduce executable paths from historical knowledge. Moreover, repeated retrieval and step-wise generation introduce substantial cost and latency, hindering real-time deployment.

In this paper, we investigate *Executable Agentic Memory* (EAM), which can serve as a persistent, structured representation of the environment interaction logic, learned from historical interactions, and can be queried at test time so that planning can be augmented by retrieval and verification rather than free-form generation. Specifically, EAM enables the agent to (1) remember the GUI as a state machine (what states exist, which actions are available, and where they lead), and (2) reason over this memory to extract an executable path that is guaranteed to stay on valid transitions.

To this end, we first propose a sample-efficient memory construction pipeline: a state-aware DFS exploration strategy that systematically covers task-relevant transitions with

---

[1]Department of Computer Science and Technology, Tsinghua University, Beijing, China [2]Sun Yat-sen University Shenzhen Campus, Shenzhen, China. Correspondence to: Ju Ren <renju@tsinghua.edu.cn>.

*Proceedings of the 43rd International Conference on Machine Learning*, Seoul, South Korea. PMLR 306, 2026. Copyright 2026 by the author(s).

minimal redundant interactions, coupled with state deduplication and action-group mining to compress frequent multi-step routines into reusable high-level actions, yielding a compact yet executable GUI logic knowledge graph. We then propose a compute-efficient retrieval mechanism: a value-guided graph search procedure in which a lightweight Q-function model steers MCTS over the constrained KG action space to rapidly select faithful high-reward paths from noisy experience; when needed, the agent can make only a single cloud call to summarize and validate the retrieved path into a grounded plan. Theoretically, we establish a bias-consistency guarantee for the learned Q-model on the critical set and derive a finite-sample complexity bound under which the value-guided MCTS recovers the optimal execution path with high probability.

We evaluate our method on the AndroidWorld, MobileMiniWob++, and DroidTask benchmarks. Results show that our framework consistently outperforms the existing baselines, surpassing the state-of-the-art UI-TARS-7B by up to $19.6\%$ while reducing token costs $6\times$ relative to GPT-4o. Our Q-guided MCTS and iterative self-training pipeline bridge the reasoning gap for small models through fine-grained credit assignment, while the action group mechanism minimizes search complexity to reach a $2.8s$ average latency. These findings demonstrate that grounding decision-making in structured knowledge graphs enables reliable, high-speed, and long-horizon planning for GUI agents.

## 2. Related Work

**GUI Agents.** Early efforts adapted foundation models (GPT-4, GPT-4o) to GUI tasks (Wen et al., 2024; Wang et al., 2023), with Zheng et al. (Zheng et al., 2024) demonstrating that GPT-4V outperforms text-based models in web scenarios. Zhang et al. (Zhang et al., 2025) augment GPT-4V with a memory module for historical actions. Subsequent work explored modular frameworks: Wang et al. (Wang et al., 2024a) integrate planning, decision, and reflection modules; Zhang et al. (Zhang et al., 2024b) propose multi-agent collaboration; and Zhu et al. (Zhu et al., 2024) design a hierarchical planner-executor architecture. However, these cloud-based frameworks incur high API costs and latency, and suffer from hallucinations due to limited GUI domain knowledge. More recent work pursues end-to-end GUI agents via parameter training. Cheng et al. (Cheng et al., 2024) train a dedicated GUI grounding model with cross-platform data, while UI-TARS (Qin et al., 2025) introduces a comprehensive pre-training to fine-tuning pipeline. To improve generalization, Luo et al. (Luo et al., 2025) and Lu et al. (Lu et al., 2025) apply rule-based RL algorithms such as GRPO (Shao et al., 2024). AutoDroid-V2 (Wen et al., 2025) fine-tunes a lightweight model to generate executable scripts in one shot. Despite these advances, on-device mod-

els ($\leq$3B) remain limited in reasoning, struggling with complex multi-step tasks.

**Knowledge-aware GUI Agents.** To mitigate hallucinations and improve adaptability, some works utilize historical memory to guide task planning. Wang et al. (Wang et al., 2024b) propose a workflow memory extracting reusable patterns from past experiences. Mobile-Agent-E (Wang et al., 2025) introduces a self-evolving framework accumulating general guidance over time. MAGNET (Sun et al., 2026) constructs dual-level memory for element grounding and workflow retrieval to handle UI drift. However, these methods rely solely on LLMs' contextual understanding without accounting for dynamic environment interactions. Another line of work focuses on reliable action generation. Auto-Droid (Wen et al., 2024) collects transition knowledge via random exploration. GUI-explorer (Xie et al., 2025) mines element functionality by analyzing GUI state changes. KG-RAG (Guan et al., 2025b) transforms UI Transition Graphs into vector databases and distills reusable actions based on intent. While these approaches improve action accuracy, path generation still relies on LLM reasoning over retrieved context rather than direct extraction from an executable state machine. Moreover, massive API calls for step-wise decision-making incur substantial costs and latency.

**LLM-based Monte Carlo Tree Search.** Inspired by AlphaGo, recent work explores guiding LLM inference with tree search to improve reasoning on structured tasks. Zhou et al. (Zhou et al., 2023) propose an LLM-MCTS framework leveraging environment feedback for decision-making, while Xie et al. (Xie et al., 2024) construct a self-learning loop using MCTS to generate preference signals for training. However, these methods require multiple LLM rollouts during simulation, limiting efficiency. More recent work employs LLMs as both policy and value models. Hao et al. (Hao et al., 2023) treat the LLM as a world model for generation and evaluation. rStar-Math (Guan et al., 2025a) trains a reward model with trajectory-level binary rewards for node scoring. ReST-MCTS* (Zhang et al., 2024a) introduces a self-trained Process Reward Model for step-wise evaluation, and Mendes et al. (Mendes & Ritter, 2025) equip the value model with look-ahead capability. Despite these advances, most methods rely on heuristic value designs without theoretical guarantees and require separate policy and value models, incurring high computational overhead.

## 3. Problem Statement

**GUI Logic Knowledge Graph.** We define the GUI Logic Knowledge Graph as a directed graph $\mathcal{G} = (\mathcal{S}, \mathcal{A}, \mathcal{E})$, where $\mathcal{S}$ denotes state nodes representing unique GUI pages, $\mathcal{A}$ denotes action nodes representing executable operations, and $\mathcal{E} \subseteq (\mathcal{S} \times \mathcal{A}) \cup (\mathcal{A} \times \mathcal{S})$ denotes edges connecting states to actions and actions to resulting states. Each state

$s \in \mathcal{S}$ contains a page description $d_s$, and each action $a \in \mathcal{A}$ is annotated with a functional description $f_a$. We denote $\mathcal{A}(s) = \{a \in \mathcal{A} : (s, a) \in \mathcal{E}\}$ as the available actions at $s$.

**Path Extraction as Finite-Horizon MDP.** Given a user instruction $x \in \mathcal{X}$, we formulate path extraction from the KG as a finite-horizon episodic MDP, $\langle \mathcal{S}, \mathcal{A}, T, R, H \rangle$. The state space $\mathcal{S}$ and action space $\mathcal{A}(s)$ are induced by the KG structure. $T$ represents a deterministic transition function where $s' = T(s, a)$ follows the KG edges. $R$ is a binary terminal reward function, where $R(s_H, x) = 1$ if terminal state $s_H$ satisfies instruction $x$, and 0 otherwise. $H$ is the horizon. At each step $t$, the agent selects $a_t \in \mathcal{A}(s_t)$ according to policy $\pi(\cdot|s_t, x)$ and transits to $s_{t+1} = T(s_t, a_t)$. The objective is to find $\pi^* = \arg\max_\pi \mathbb{E}_{a_t \sim \pi}[R(s_H, x)]$ that identifies a successful path $\tau^*$ for instruction $x$.

# 4. Methodology

In this section, we introduce our proposed agentic memory system which comprises two main components: 1) Offline Knowledge Graph Construction, which autonomously explores the GUI environment to collect transition data and builds a structured knowledge graph $\mathcal{G}$; and 2) Online Knowledge-Augmented Reasoning, which leverages the constructed KG to extract faithful execution paths via Q-model guided MCTS. An overview of the framework is presented in Fig. 1. We elaborate on each component in the following subsections.

## 4.1. Offline Knowledge Graph Construction

The offline stage aims to construct a comprehensive GUI Logic Knowledge Graph $\mathcal{G} = (\mathcal{S}, \mathcal{A}, \mathcal{E})$ that captures both the structural logic and semantic knowledge of the target GUI environment. This process consists of three key components: autonomous exploration for trajectory collection, transition-aware knowledge mining for graph construction, and action group mining for efficient high-level guidance.

**Autonomous Exploration.** The core of our offline stage lies in task-oriented autonomous exploration that systematically discovers GUI states and transitions contributing to task completion. We propose an element-grounded hierarchical exploration based on depth-first search (DFS). Given a task goal $g$, we extract *Exploration Anchors* from the current GUI state—interactable elements serving as structural primitives for sub-goal generation. The MLLM uses these anchors to generate up to $k$ candidate sub-goals ranked by their likelihood of progressing toward $g$. At each depth, the agent evaluates progress and determines one of three outcomes: (1) CONTINUE—the sub-goal was achieved but $g$ requires further operations; (2) BACKTRACK—the current state deviates from the path toward $g$; (3) COMPLETE—the task goal $g$ is achieved. This DFS-based design ensures

comprehensive coverage of task-relevant transitions (up to $O(k^d)$ distinct trajectories) while the collected trajectories naturally form a prefix tree structure that can be seamlessly transformed into the knowledge graph $\mathcal{G}$.

**Transition-aware Knowledge Mining.** The knowledge construction process builds a structured KG from collected exploration trajectories. Let $\xi = \langle s_0, a_0, s_1, a_1, \ldots, s_n \rangle$ denote an interaction trajectory. Following (Wen et al., 2025; Xie et al., 2025), we extract transition-aware GUI knowledge by analyzing consecutive transitions to construct the graph structure and enrich semantic attributes.

*1) Graph Structure Construction:* The KG is constructed as a Directed Acyclic Graph (DAG) where state nodes and action nodes alternate, with each trajectory incrementally merged into the KG. The key challenge lies in accurately mapping new trajectories to the existing state space. To this end, we design a state-aware deduplication mechanism featuring dual-layer filtering: *(i) Coarse Filtering*—each new state is encoded by an embedding model and matched against existing states via similarity retrieval; *(ii) Fine-grained Filtering*—candidate duplicates are verified by a Vision-Language Model for rigorous semantic comparison. For duplicate states, we further perform element-level deduplication via IoU of bounding boxes, effectively connecting discrete exploration trajectories into a cohesive graph.

*2) Semantic Knowledge Enrichment:* Once the topological structure is established, we enrich the graph with semantic attributes derived from state transitions. The knowledge mining process is formalized as:

$$\mathcal{G} \leftarrow \mathcal{G} \oplus \mathcal{F}_{\text{extract}}(s_t, a_t, s_{t+1}) \tag{1}$$

where $\mathcal{F}_{\text{extract}} : (s_t, a_t, s_{t+1}) \mapsto (d_{s_t}, d_{s_{t+1}}, f_{a_t})$ generates page descriptions and action functional descriptions from the state transition, and $\oplus$ denotes the merge operator that continuously updates the extracted knowledge into $\mathcal{G}$.

**Action Group Mining.** Beyond atomic actions, real-world GUI tasks often involve recurring multi-step action patterns. While recent works extract high-level actions from trajectories (Jiang et al., 2025; Wang et al., 2025), they rely heavily on LLMs to summarize these groups, suffering from poor cross-task generalizability and high computational cost.

To address these limitations, we propose a statistical approach inspired by Byte Pair Encoding (BPE). We conceptualize the KG as a "path heatmap," where high-frequency action subsequences represent high-value generalizable skills. Formally, let $\mathcal{V} = \{a_1, a_2, \ldots, a_M\}$ denote the initial vocabulary of atomic actions, and let $\mathcal{P} = \{\tau_1, \tau_2, \ldots, \tau_K\}$ denote the corpus of all historical paths in the KG, where each path $\tau = (a_{i_1}, a_{i_2}, \ldots, a_{i_L})$ is a sequence of atomic actions.

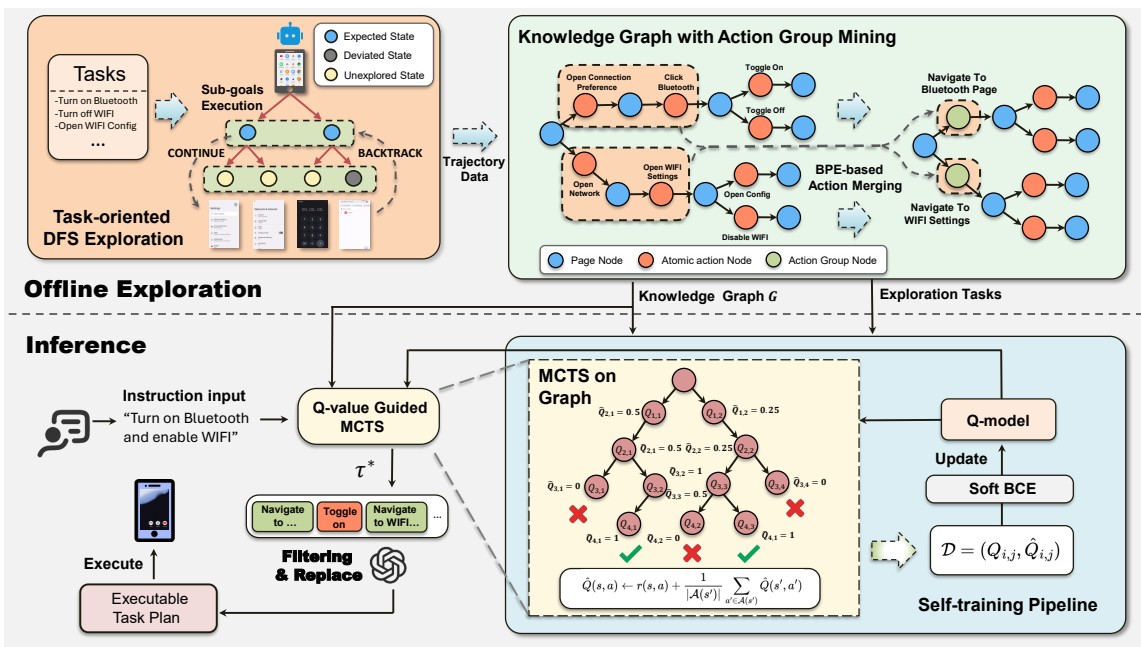

*Figure 1.* Overview of Executable Agentic Memory (EAM). It comprises offline automatic memory construction and inference-time executable memory reuse guided by a trained Q-model.

The mining process proceeds iteratively. At each iteration $j$, we compute the frequency of all adjacent action pairs and identify the most frequent pair:

$$(a^*, a'^*) = \arg \max_{(a,a') \in \mathcal{V} \times \mathcal{V}} \sum_{\tau \in \mathcal{P}} \text{count}((a, a'), \tau) \quad (2)$$

where $\text{count}((a, a'), \tau)$ denotes the number of occurrences of the adjacent pair $(a, a')$ in path $\tau$. If the maximum frequency exceeds a predefined threshold $\delta_f$, we merge the pair into a new action group and update the vocabulary:

$$a_{\text{new}}^{(j)} = a^* \circ a'^*, \quad \mathcal{V} \leftarrow \mathcal{V} \cup \{a_{\text{new}}^{(j)}\}. \quad (3)$$

The corpus $\mathcal{P}$ is then updated by replacing all occurrences of $(a^*, a'^*)$ with $a_{\text{new}}^{(j)}$. This process iterates until the frequency of the most common pair falls below $\delta_f$. The mined action groups are integrated into the KG as high-level action nodes, extending the action space from atomic operations to multi-step reusable skills.

### 4.2. Online Knowledge-Augmented Path Extraction

Given a user instruction $x$, extracting an executable path from the KG can be formulated as the finite-horizon MDP defined in Section 3. This MDP features deterministic transitions, binary terminal rewards, and a relatively small state-action space constrained by the KG structure. Such a tabular

setting differs fundamentally from classical agentic RL scenarios, which typically involve complex reward structures and vocabulary-scale action spaces. Due to this structural mismatch, directly employing mainstream GRPO-style RL frameworks (Shao et al., 2024; Feng et al., 2025; Jin et al., 2025) is suboptimal, as they easily suffer from entropy collapse and introduce unnecessary computational overhead.

To address this challenge, we introduce a *path navigating agent* that leverages Monte Carlo Tree Search (MCTS) guided by a lightweight Q-model to extract executable paths from the KG. Unlike generative agents that map generated tokens into the graph, our agent explicitly operates on the graph topology and treats reasoning as planning over discrete states and actions. This design offers three key advantages. First, by constraining the action space to valid edges in $\mathcal{G}$, the agent naturally decouples graph reasoning from semantic generation and treats the KG as a rigorous state machine. Second, the agent automatically generates step-level Q-value annotations through MCTS rollouts, which obviates the need for human-labeled training data. Third, instead of fine-tuning a generative model over a vast vocabulary, the agent relies on a compact Q-model to predict scalar values, which significantly reduces computational cost.

Our path navigating agent consists of three components: Q-model guided MCTS framework, random policy valuation

for node evaluation, and a self-training pipeline for iterative model refinement.

**Q-model Guided MCTS.** The agent performs tree search on the KG starting from a root node $h_0 = (x, s_0)$, which encodes the instruction $x$ and initial state $s_0$. Each node $h_t = (s_t, a_t)$ in the search tree corresponds to a state-action pair. The search proceeds through four MCTS phases:

*1) Selection:* The agent traverses the tree by selecting child nodes according to the UCT criterion until reaching a leaf node:

$$\text{UCT}(s, a) = Q(s, a) + c\sqrt{\frac{\ln N(s)}{N(s, a)}} \qquad (4)$$

where $Q(s, a)$ is the estimated Q-value, $N(s, a)$ the visit count, and $c$ the exploration constant.

*2) Expansion:* Upon reaching a non-terminal leaf state $s_l$, the agent expands all available actions $a \in \mathcal{A}(s_l)$ as child nodes.

*3) Evaluation:* Unlike standard MCTS with random roll-outs, the agent queries its Q-model to initialize Q-values: $Q(s_l, a) \leftarrow Q_\theta(s_l, a)$, where $Q_\theta(s, a) \in (0, 1)$ predicts the task success probability.

*4) Back-propagation:* The agent propagates Q-values back to the root, updating visit counts and Q-estimates along the path via incremental averaging.

$$N(s_t, a_t) \leftarrow N(s_t, a_t) + 1$$
$$Q(s_t, a_t) \leftarrow Q(s_t, a_t) + \frac{Q(s_l, a_l) - Q(s_t, a_t)}{N(s_t, a_t)}. \qquad (5)$$

After $M$ iterations, the top-$K$ paths with the highest mean Q-values are extracted and processed by a cloud-based LLM for one-time filtering and parameter replacement into the final executable plan.

**Random Policy Valuation.** To effectively guide the search, the agent's Q-model should not only identify superior actions but also quantify the likelihood of success after selecting each action. Most existing MCTS frameworks employ Outcome Reward Models (ORM) that assign binary values (Cobbe et al., 2021). Such coarse signals overlook the nuanced differences among intermediate steps.

Instead, we define the Q-value as the expected success probability under a uniform random policy $\pi_u(a|s) = 1/|\mathcal{A}(s)|$. This value can be computed via the Bellman equation:

$$Q^{\pi_u}(s, a) = r(s, a) + \frac{1}{|\mathcal{A}(s')|} \sum_{a' \in \mathcal{A}(s')} Q^{\pi_u}(s', a'). \qquad (6)$$

$s' = T(s, a)$ is the successor state, and $r(s, a) \in \{0, 1\}$ is the terminal reward. The value $Q^{\pi_u}(s, a)$ represents the probability of reaching a successful terminal state when

starting from $(s, a)$ and acting uniformly at random thereafter. When $Q^{\pi_u}(s, a) = 0$, no feasible path exists from $(s, a)$ to success, whereas higher values indicate greater likelihood of task completion. Crucially, recent theoretical results have shown that acting greedily with respect to $Q^{\pi_u}$ achieves optimality in finite-horizon deterministic MDPs with binary rewards (He et al., 2025; Laidlaw et al., 2023), which aligns precisely with our KG setting.

**Self-Training Pipeline.** We train the agent's Q-model $Q_\theta$ through an iterative self-training procedure consisting of an initialization stage and a refinement stage.

*1) Initialization:* Directly deploying an untrained Q-model leads to random exploration and severe label imbalance, as the search predominantly encounters dead-end nodes with zero Q-values. To address this cold-start problem, we initialize $Q_\theta$ using preference learning on an existing GUI dataset. For each step $t$ along an expert trajectory, we construct preference pairs with the expert action $a_t^+$ as positive and a randomly sampled $a_t^- \in \mathcal{A}(s_t) \setminus \{a_t^+\}$ as negative. The agent is trained with a pairwise ranking loss based on the Bradley-Terry model:

$$\mathcal{L}_{\text{init}}(\theta) = -\mathbb{E}_{(\tau_t^+, \tau_t^-) \sim \mathcal{D}_{\text{init}}} \left[ \mathcal{R}(\tau_t^+, \tau_t^-) \right] \qquad (7)$$

where $\mathcal{R}(\tau_t^+, \tau_t^-) = \log \sigma(Q_\theta(s_t, a_t^+) - Q_\theta(s_t, a_t^-))$.

*2) Iterative Refinement:* After initialization, the agent iteratively refines its Q-model using self-generated data. In each round, the agent samples instructions and executes MCTS guided by the current $Q_\theta$ to construct search trees, then computes target Q-values via bottom-up Bellman backup:

$$\hat{Q}(s, a) \leftarrow r(s, a) + \frac{1}{|\mathcal{A}(s')|} \sum_{a' \in \mathcal{A}(s')} \hat{Q}(s', a'). \qquad (8)$$

Since Q-values represent probabilities in $[0, 1]$, we formulate the optimization as binary classification with soft labels:

$$\mathcal{L}_{\text{update}}(\theta) = -\mathbb{E}_{(s,a) \sim \mathcal{T}} \left[ \hat{Q} \log p_\theta + (1 - \hat{Q}) \log(1 - p_\theta) \right] \qquad (9)$$

where $p_\theta = \sigma(Q_\theta(s, a))$ and $\mathcal{T}$ denotes state-action pairs from the search trees. Through this iterative process, the agent progressively improves its ability to identify promising paths within the KG.

## 5. Theoretical Analysis

In this section, we provide theoretical guarantees for the proposed Q-model guided MCTS framework. We first formalize the problem setting, then present our two main results: (1) a bias consistency guarantee ensuring the learned Q-model is close to $Q^{\pi_u}$ on critical states, and (2) a sample complexity bound for extracting the optimal path.

## Algorithm 1 Self-Training for Path-Navigating Agents

**Input:** Instruction dataset $\mathcal{D}$, initialization dataset $\mathcal{D}_{\text{init}}$, knowledge graph $\mathcal{G}$
**Output:** Trained Q-model $Q_\theta$
▷ Model Initialization
Construct preference pairs $(\tau_t^+, \tau_t^-)$ from $\mathcal{D}_{\text{init}}$
Initialize $Q_\theta$ by minimizing ranking loss $\mathcal{L}_{\text{init}}$ (Eq. 7)
▷ Iterative Refinement via MCTS
**for** each round $r = 1, 2, \ldots, R$ **do**
    Sample instruction batch $\mathcal{B}$ from $\mathcal{D}$
    $\mathcal{T} \leftarrow \emptyset$
    **for** each instruction $x \in \mathcal{B}$ **do**
        Execute MCTS guided by $Q_\theta$ on $\mathcal{G}$ to construct search tree
        Compute $\hat{Q}(s, a)$ for all nodes (Eq. 8)
        $\mathcal{T} \leftarrow \mathcal{T} \cup \{(s, a, \hat{Q}(s, a))\}$
    **end for**
    Update $Q_\theta$ by minimizing $\mathcal{L}_{\text{update}}$ (Eq. 9)
**end for**
**Return:** $Q_\theta$

### 5.1. Problem Setting

We analyze path extraction on $\mathcal{G}$ under the MDP formulation from Section 3. Proposition 5.1 formalizes the optimality guarantee of the greedy policy with respect to $Q^{\pi_u}$.

**Proposition 5.1** (Optimality of Greedy Policy (He et al., 2025)). *Consider the KG-induced MDP with deterministic transitions, tree-structured state space, and binary terminal rewards $r \in \{0, 1\}$. Let $\pi_u$ be the uniform policy and $Q^{\pi_u}$ its corresponding Q-function. Define the greedy policy $\pi_{greedy}(s) = \arg\max_{a \in \mathcal{A}(s)} Q^{\pi_u}(s, a)$. Then $\pi_{greedy}$ is optimal.*

Let $\tau^* = (s_0^*, a_0^*, \ldots, s_{H-1}^*, a_{H-1}^*)$ denote an optimal path induced by $\pi_{\text{greedy}}$. Define the *critical set* $\mathcal{C}$ as the collection of state-actions that must be ranked correctly to recover $\tau^*$:

$$\mathcal{C} = \bigcup_{t=0}^{H-1} \{(s_t^*, a) : a \in \mathcal{A}(s_t^*)\}. \tag{10}$$

Let $|\mathcal{C}| = H \cdot \max_s |\mathcal{A}(s)|$. Define the minimum action gap along the optimal path:

$$\Delta_{\min}^* := \min_{t \in \{0, \ldots, H-1\}} \left[ Q^{\pi_u}(s_t^*, a_t^*) - \max_{a \neq a_t^*} Q^{\pi_u}(s_t^*, a) \right]. \tag{11}$$

We train $Q_\theta$ using target values computed via uniform Bellman backup (Eq. 8) and perform MCTS at inference to extract the optimal path.

### 5.2. Main Results

Next, we give the bias consistency guarantee on the critical set.

**Theorem 5.2** (Bias Consistency on $\mathcal{C}$). *With probability at least $1 - \delta$, the learned predictor $Q_\theta$ satisfies*

$$\|Q_\theta - Q^{\pi_u}\|_{2,\rho_{\mathcal{C}}} \leq \epsilon_{\text{bias}}(m, \delta) \tag{12}$$

*where*

$$\epsilon_{\text{bias}}(m, \delta) := \sqrt{\frac{1}{2} \left( \epsilon_{\text{approx}} + 2\varepsilon_{\text{gen}}(m, \delta/2) + \epsilon_{\text{opt}} \right)}. \tag{13}$$

$\epsilon_{\text{approx}}$ *is the in-class approximation error, $\varepsilon_{\text{gen}}(m, \delta)$ is the generalization error depending on $m$ and Rademacher complexity, and $\epsilon_{\text{opt}}$ is the optimization error.*

Theorem 5.2 shows that the proxy error decreases as training samples increase. This bias bound directly controls the accuracy of MCTS node evaluation: when $\epsilon_{\text{bias}} < \Delta_{\min}^*/2$, the learned Q-model preserves correct action rankings on the critical set (see Appendix A.1 for details).

Our second main result establishes the sample complexity for optimal path extraction.

**Theorem 5.3** (Sample Complexity for Optimal Path Extraction). *Suppose $\epsilon_{\text{bias}}(m, \delta/2) < \Delta_{\min}^*/2$. Let $\Delta_{\text{eff}} := \Delta_{\min}^* - 2\epsilon_{\text{bias}} > 0$ and $K = \max_s |\mathcal{A}(s)|$. Then for the greedy path $\hat{a}_t = \arg\max_a \bar{Q}_n(s_t^*, a)$ to coincide with $\tau^*$ with probability at least $1 - \delta$, the number of MCTS simulations per node must satisfy*

$$n \geq \frac{32(K-1)c^2 \ln(Hn/\delta)}{\Delta_{\text{eff}}^2} + 2(K-1)\left(2N_0 + \frac{\pi^2}{3}\right) \tag{14}$$

*yielding total complexity $N_{\text{total}} = O\left(\frac{HKc^2 \ln(Hn/\delta)}{(\Delta_{\min}^* - 2\epsilon_{\text{bias}})^2}\right) + O(HKN_0)$, where $c$ is the UCT exploration constant and $N_0$ is a burn-in threshold.*

Theorem 5.3 shows that the simulation complexity scales polynomially with horizon $H$, branching factor $K$, and inversely with the squared effective action gap. This provides a theoretical foundation for the efficiency of our approach: as the Q-model improves (reducing $\epsilon_{\text{bias}}$), fewer MCTS simulations are needed to recover the optimal path. Complete proofs are provided in Appendix A.2.

## 6. Experiment

In this section, we will present the results of our empirical study to answer the following question:

- How does our proposed method perform on standard GUI benchmarks compared to both on-device and cloud-based baselines in terms of success rate and efficiency?

| Method | Base Model | Input | AndroidWorld (%) | MobileMiniWob++ (%) | DroidTask (%) |
|---|---|---|---|---|---|
| GPT-4o | GPT-4o | SoM | 34.5 | 56.5 | 57.0 |
| Qwen 2.5-VL-3B | Qwen 2.5-VL-3B | SoM | 2.6 | 32.6 | 13.3 |
| UI-TARS-2B | UI-TARS-2B | screen | 6.9 | 31.5 | 34.8 |
| UI-TARS-7B | UI-TARS-7B | screen | 33.0 | 53.3 | 55.0 |
| M3A | GPT-4o | SoM | 40.5 | 68.5 | 72.2 |
| AutoDroid-V2 | Llama-3-8B-ft | SoM | 26.0 | 53.3 | 54.4 |
| AppAgentX | GPT-4o | SoM | 62.5 | 72.8 | 88.6 |
| GUI-Explorer | GPT-4o | SoM | 47.4 | 80.4 | 88.0 |
| EAM (Ours) | GPT-4o, Qwen2.5-3B-instruct-ft | SoM | **52.6** | **76.1** | **86.1** |

*Table 1.* Success rate (%) comparison between our method and baselines on AndroidWorld, MobileMiniWob++, and DroidTask benchmarks. "SoM" refers to Set-of-Mark prompting, which utilizes the bounding boxes recorded in the accessibility tree to annotate UI elements with numerical labels in screenshots. All results are averaged over three independent runs.

| Method | Latency (s) | API Tokens Cost (K) |
|---|---|---|
| GPT-4o | 9.3 | 50.8 |
| Qwen2.5-VL-3B | 7.7 | - |
| UI-TARS-2B | 6.0 | - |
| UI-TARS-7B | 8.8 | 32.7 |
| M3A | 16.9 | 62.6 |
| AutoDroid-V2 | 2.1 | - |
| AppAgentX | 16 | 6.2 |
| GUI-Explorer | 66.4 | 73.1 |
| EAM (Ours) | **2.8** | **8.3** |

*Table 2.* Efficiency comparison between EAM and baselines in terms of latency and token cost. "Latency (s)" denotes the average execution time per step. "API Tokens Cost (K)" indicates the total token consumption (in thousands) per step for LLM API calls. "-" indicates that the method uses locally deployed models without API calls.

- Does our self-training pipeline enable stable iterative performance improvements and exhibit theoretically expected properties?

- How do the various components in our method affect performance, and does the trained Q-model demonstrate cross-environment generalization?

### 6.1. Experimental Setup

**Benchmarks.** We evaluate the effectiveness and efficiency our method on three benchmarks: AndroidWorld (Rawles et al., 2024) (116 tasks across 20 real-world apps), MobileMiniWob++ (Rawles et al., 2024) (92 web tasks), and DroidTask (Wen et al., 2024) (158 tasks across 13 apps).

**Baselines.** For on-device agents, we consider four baselines: 1) Qwen2.5-VL-3B, the vanilla VLM for on-device deployment; 2) UI-TARS-2B (Qin et al., 2025), the lightweight SFT version of UI-TARS-7B; 3) UI-TARS-7B (Qin et al., 2025), a SOTA GUI agent model; 4) AutoDroid-V2 (Wen et al., 2025), a code-generation agent fine-tuned on Llama-

3-8B that produces executable scripts for one-shot task execution. For cloud-based and knowledge-enhanced agents, we consider: 1) GPT-4o, the base VLM for cloud-based agents; 2) M3A (Rawles et al., 2024), a SOTA ReAct-based agent framework; 3) AppAgentX (Jiang et al., 2025), which extracts and reuses high-level actions from GUI transitions for task guidance; 4) GUI-Explorer (Xie et al., 2025), an exploration-augmented framework that collects trajectories, extracts element-wise knowledge, and uses RAG for decision-making.

**Implementation.** Our framework is implemented as a plug-and-play module built on UI-TARS-2B, which serves as a local action executor following memory-grounded planning. We use GPT-4o for task-oriented exploration and knowledge mining. The knowledge base is constructed with Neo4j for app-wise knowledge graphs and Pinecone for screenshot embeddings. We fine-tune Qwen2.5-Instruct for path extraction with three model sizes: 0.5B, 1.5B, and 3B. For Q-value estimation, we append a value head to output scalar predictions. The self-training pipeline runs for four rounds. All training is conducted on $4\times$A800-80GB GPUs, and inference experiments are performed on a single RTX 4090-16GB to simulate on-device deployment.

### 6.2. Experimental Results

**Comparative results.** Table 1 reports success rates on the three benchmarks. Our method achieves **52.6%** on AndroidWorld, **76.1%** on MobileMiniWob++, and **86.1%** on DroidTask, surpassing all on-device baselines by significant margins (+19.6, +22.8, and +31.1, respectively). Notably, despite utilizing a 3B model for path extraction, our method substantially outperforms GPT-4o based M3A (+7.6, +13.9, and +29.1) and achieves performance comparable to knowledge-enhanced agents like AppAgentX and GUI-Explorer. These gains indicate that grounding decision-making in a structured knowledge graph effectively bridges the reasoning gap between small language models and frontier LLMs. Table 2 demonstrates that our approach achieves

an average latency of **2.8 s** and token cost of **8.3K** per step. This efficiency stems from our plan-then-execute framework: unlike cloud-based agents requiring massive iterative API calls, our method necessitates only a single API call to filter the extracted paths, reducing token cost by approximately $6\times$ compared to GPT-4o (50.8K). While AutoDroid-V2 also adopts plan-then-execute to achieve low latency (2.1 s), its performance suffers due to a lack of rigorous knowledge guidance during inference.

**Effect of dense Q-value guided MCTS.** We analyze the two key factors behind path extraction: the value supervision used to train the path-navigating model and the search strategy used at inference. For value supervision, we compare an untrained base model, preference-based initialization, binary outcome supervision, and our soft Q-value supervision. All variants use the same 3B backbone and MCTS configuration ($N = 50$, $c = 10$). As shown in Fig. 2a–2b, preference-based initialization improves over the untrained model, while binary supervision remains limited by coarse path-level labels. Q-value training performs best, showing that dense Bellman-backed targets provide more effective intermediate credit assignment.

We further compare Greedy selection, Best-of-$N$ sampling, and MCTS using the trained Q-model. Fig. 2c–2d show that MCTS consistently outperforms the alternatives on both AndroidWorld and DroidTask. Greedy selection is efficient but prone to early mistakes, while Best-of-$N$ samples multiple paths without structured value propagation. MCTS better balances exploration and exploitation, making path extraction more robust under noisy value estimates.

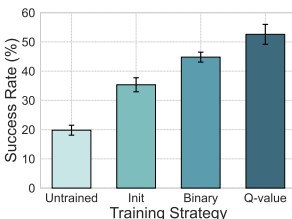

*(a)* Q-value design on Android-World

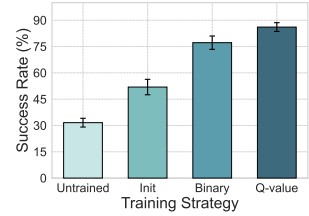

*(b)* Q-value design on Droid-Task

*(c)* Search strategy on Android-World

*(d)* Search strategy on Droid-Task

*Figure 2.* Effect of dense Q-value guided MCTS. Panels (a-b) compare different training targets for the path-navigating model, while panels (c-d) compare path extraction strategies using the trained Q-model.

**Substantial improvement through self-training.** To answer Q.2, we analyze performance gains through iterative self-training. As shown in Fig. 3a–3b, all three model sizes (0.5B, 1.5B, and 3B) exhibit consistent improvement on both AndroidWorld and DroidTask as self-training rounds increase. Notably, the 3B model achieves the most substantial gains. Fig. 3c–3d further shows the value gap between optimal and suboptimal actions on critical paths: as self-training progresses, the model develops more accurate value estimates, enabling better separation of optimal actions. The initially negative or near-zero margins in early rounds indicate that the untrained model struggles to distinguish optimal actions, while later rounds show increasingly positive margins, demonstrating improved discriminative ability. This aligns with our theoretical analysis bounding path extraction ability by sample size and value estimation quality: as $\epsilon_{\text{bias}}$ decreases through training, the effective action gap $\Delta_{\text{eff}} = \Delta_{\text{min}}^* - 2\epsilon_{\text{bias}}$ increases, requiring fewer MCTS simulations to recover optimal paths (Theorem 5.3).

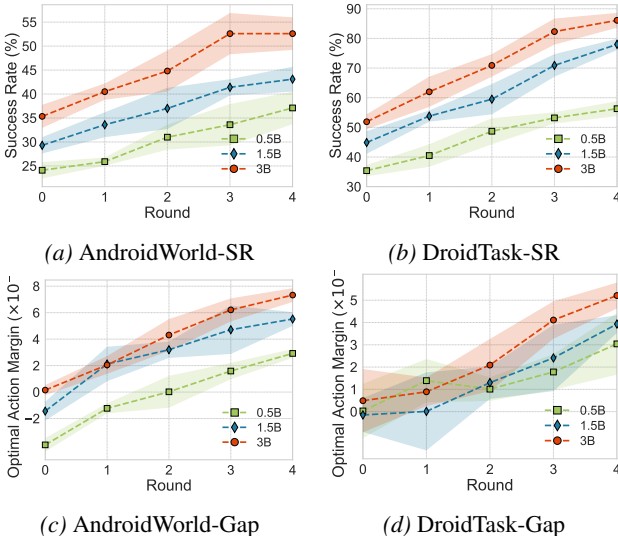

*(a)* AndroidWorld-SR

*(b)* DroidTask-SR

*(c)* AndroidWorld-Gap

*(d)* DroidTask-Gap

*Figure 3.* Self-training dynamics: (a-b) Success rate and (c-d) optimal action margin across rounds.

**Offline construction cost.** Table 3 reports the one-time cost of building EAM on AndroidWorld in terms of both wall-clock time and token consumption. Building an app-level memory takes about 80 minutes per app, where DFS exploration dominates the cost and KG construction adds a smaller post-processing overhead. The token cost follows a similar pattern. Although offline cost is higher than a single online inference step, it is amortized across future tasks in the same app and becomes negligible as the number of online tasks grows.

**Component ablation and generalization.** Fig. 4 quantifies the contribution of each component. Starting from the Baseline (UI-TARS-2B without KG guidance), intro-

| | DFS Exploration | | KG Construction | |
|---|---|---|---|---|
| | Time (s) | Tokens (K) | Time (s) | Tokens (K) |
| Per App | 3768.4 | 1141.4 | 1019.2 | 214.8 |
| Per Task | 649.7 | 196.8 | 175.7 | 37.0 |

*Table 3.* Average offline construction cost on AndroidWorld. DFS exploration denotes task-oriented environment traversal, while KG construction denotes graph construction including state deduplication, transition mining, and action group mining.

ducing the knowledge graph with an initialization-trained model (+KB w/ init model) yields substantial gains, demonstrating the value of structured knowledge. Training the Q-model in-environment (+In-env Model) achieves the best performance, confirming that domain-specific training further refines value estimates. Notably, replacing with a cross-environment trained model (+Cross-env Model, trained on AndroidWorld) also improves performance on DroidTask and MobileMiniWob++, despite never seeing these environments during training. This addresses Q.3: the learned value estimation captures transferable knowledge about GUI navigation patterns, enabling reliable path extraction in unseen scenarios. While in-environment training remains optimal, cross-environment results suggest a well-trained Q-model can serve as strong initialization for new environments.

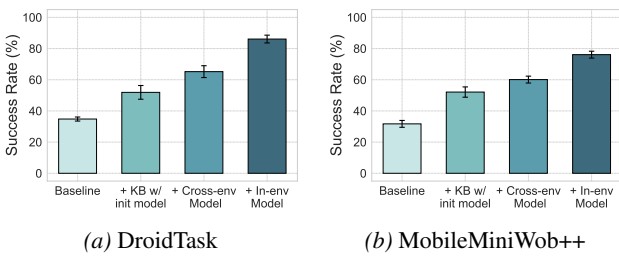

*(a)* DroidTask      *(b)* MobileMiniWob++

*Figure 4.* Ablation study on agentic memory with different model: cross-environment vs. in-environment trained models.

**Action groups.** We construct ablation experiments on action groups. Fig. 5a–5b present the length distribution of actions in the KGs. Beyond atomic actions, our BPE-based merging mechanism constructs action groups encapsulating multi-step sequences, with lengths following a long-tail distribution. AndroidWorld exhibits more long-sequence groups due to higher task complexity. These action groups merge multiple frequently co-occurring atomic action nodes into a single high-level action node, thereby reducing the depth and branching complexity of the KG. As a result, subsequent MCTS can search over a more compact action space while still preserving executable multi-step behavior. Fig. 5c–5d show that action groups yield significant improvements in both success rate and latency, especially on AndroidWorld. This is because AndroidWorld tasks involve longer operation chains and more heterogeneous app states,

making search depth reduction particularly beneficial.

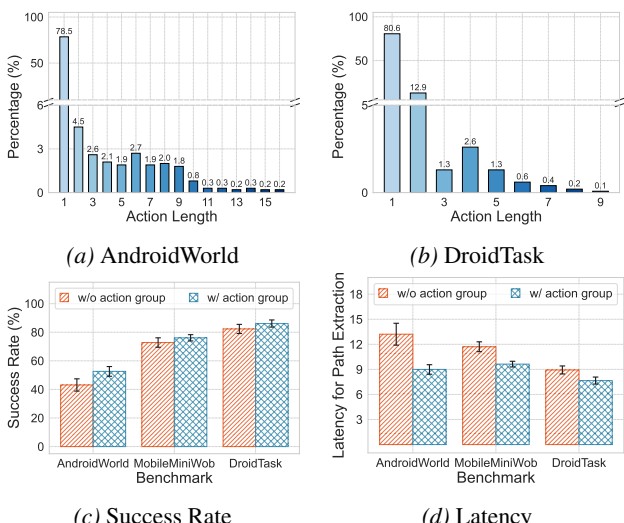

*(a)* AndroidWorld      *(b)* DroidTask

*(c)* Success Rate      *(d)* Latency

*Figure 5.* Ablation study on action groups: (a-b) Distribution action sequence lengths in KGs and (c-d) Effect of action group on performance and efficiency.

More experimental details, including analyses of training loss curves, action groups, model size, and MCTS hyperparameters, can be found in the Appendix B.

## 7. Conclusion

This paper presents Executable Agentic Memory (EAM), a structured knowledge graph that shifts GUI planning from free-form generation to a robust retrieval-and-execution process. Unlike prior knowledge-augmented approaches that rely on LLM reasoning over retrieved context, EAM enables agents to directly extract executable paths guaranteed to stay on valid transitions. We propose a sample-efficient memory construction pipeline and a value-guided MCTS framework with theoretical guarantees for reliable path extraction. Experimental results across three benchmarks demonstrate significant improvements in both success rate and efficiency. These findings show that treating the KG as an executable state machine, rather than a retrieval source for in-context injection, enables reliable, efficient, and long-horizon GUI automation.

**Limitations.** Our current framework assumes a relatively static UI environment. When applications undergo significant updates, the knowledge graph may become outdated. Developing efficient incremental evolution mechanisms for the knowledge graph to adapt to the frequently updating environments remains an important direction for future work.

# Acknowledgements

This research was supported in part by the National Natural Science Foundation of China under Grants 62572496 and 62432004, the Fundamental and Interdisciplinary Disciplines Breakthrough Plan of the Ministry of Education of China under Grant No. JYB2025XDXM122, the Guangdong Natural Science Foundation under Grant 2026A1515011265, the Shenzhen Science and Technology Program under Grant JCYJ20250604175500001, the Young Elite Scientist Sponsorship Program by CAST under Contract ZB2025-218, and a grant from the Guoqiang Institute, Tsinghua University.

# Impact Statement

This paper presents Executable Agentic Memory (EAM), a framework for improving the reliability and efficiency of long-horizon GUI automation agents. If deployed responsibly, such systems could reduce repetitive digital work, improve productivity, and support accessibility by helping users complete multi-step tasks in mobile and web applications. At the same time, GUI automation can be misused for harmful purposes, including unauthorized actions, automated abuse of online services, and privacy-invasive data collection. Our approach also raises privacy and security considerations because building and using agent memory may involve storing interaction traces or screenshots that could contain sensitive information. To mitigate these risks, we recommend deployments incorporate explicit user consent, least-privilege access, careful handling and redaction of stored artifacts, and monitoring/auditing of automated actions. We encourage future work on safety constraints for high-risk operations and privacy-preserving mechanisms for agent memory.

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

# A. Theoretical Proof

This appendix provides complete proofs of all theoretical results stated in Section 5. We organize the proofs following the structure of the main text.

## A.1. Proof of Theorem 5.2

Classical MCTS with terminal rollouts produces node estimates that concentrate around their own expectations. In contrast, we use a learned value model $Q_\theta$ to guide tree search, which introduces additional bias that must be controlled. A necessary condition for reliable navigation is that the proxy $Q_\theta$ is consistent with the reference value $Q^{\pi_u}$ on the *critical set* $\mathcal{C}$, so that action rankings on the critical path are preserved. Thus, we first aim to control the *bias* induced by using a learned value model. Specifically, for any critical pair $(s, a) \in \mathcal{C}$, we aim to bound

$$\left| \mathbb{E}\big[\bar{Q}_N(s, a)\big] - Q^{\pi_u}(s, a) \right|, \tag{15}$$

$\bar{Q}_N(s, a)$ denotes the MCTS estimate produced at inference time after $N$ visits. The expectation is taken over the internal randomness of MCTS. This bias control provides the interface to standard UCT/MCTS finite-sample analysis, where $\bar{Q}_N(s, a)$ concentrates around its mean. Formally, for any $(s, a) \in \mathcal{C}$,

$$\begin{aligned} \left| \bar{Q}_N(s, a) - Q^{\pi_u}(s, a) \right| \leq & \left| \bar{Q}_N(s, a) - \mathbb{E}\big[\bar{Q}_N(s, a)\big] \right| \\ & + \left| \mathbb{E}\big[\bar{Q}_N(s, a)\big] - Q^{\pi_u}(s, a) \right|. \end{aligned} \tag{16}$$

The first term is the standard finite-sample deviation of MCTS and will be bounded by standard UCT/MCTS concentration results. In this section, we focus on controlling the second term, which captures the bias induced by using a learned value model. Such proxy bias can be further decomposed into two components:

- **Target error (estimation noise).** The target $\hat{Q}$ deviates from $Q^{\pi_u}$ due to finite terminal rollouts.

- **Learning/generalization error.** The learned predictor $Q_\theta$ deviates from the target mapping due to finite training samples and optimization error.

**Target Estimation:** Let $Z(s, a) \in [0, 1]$ be the discounted terminal return obtained by rolling out from $(s, a)$ to termination under $\pi_u$. Then

$$\mathbb{E}[Z(s, a) \mid s, a] = Q^{\pi_u}(s, a). \tag{17}$$

**Lemma A.1** (Unbiasedness of Bellman Backup)**.** *Let the uniform-policy Bellman operator $\mathcal{T}$ for deterministic transitions be*

$$(\mathcal{T}\hat{Q})(s, a) = r(s, a) + \frac{1}{|\mathcal{A}(s')|} \sum_{a' \in \mathcal{A}(s')} \hat{Q}(s', a') \tag{18}$$

*where $s' = T(s, a)$. If each child estimate is unbiased for $Q^{\pi_u}$, then the backed-up value is also unbiased.*

*Proof.* Let $s' = T(s, a)$ denote the successor state under deterministic transition. Suppose for all $a' \in \mathcal{A}(s')$, the child estimates satisfy

$$\mathbb{E}[\hat{Q}(s', a')] = Q^{\pi_u}(s', a'). \tag{19}$$

The backed-up value is

$$\hat{Q}(s, a) = r(s, a) + \frac{1}{|\mathcal{A}|} \sum_{a' \in \mathcal{A}(s')} \hat{Q}(s', a'). \tag{20}$$

Taking expectations and using linearity:

$$\mathbb{E}[\hat{Q}(s,a)] = r(s,a) + \gamma \frac{1}{|\mathcal{A}|} \sum_{a' \in \mathcal{A}(s')} \mathbb{E}[\hat{Q}(s',a')] \tag{21}$$

$$= r(s,a) + \gamma \frac{1}{|\mathcal{A}|} \sum_{a' \in \mathcal{A}(s')} Q^{\pi_u}(s',a') \tag{22}$$

$$= r(s,a) + \gamma \mathbb{E}_{a' \sim \pi_u}[Q^{\pi_u}(s',a')] \tag{23}$$

$$= (\mathcal{T}Q^{\pi_u})(s,a) \tag{24}$$

$$= Q^{\pi_u}(s,a), \tag{25}$$

where the last equality follows from the Bellman fixed-point equation for $Q^{\pi_u}$ under policy $\pi_u$. $\qquad \square$

Given the target values of leaf nodes obtained by rolling out under $\pi_u$, Lemma A.1 propagate Unbiasedness to the whole tree. We can further bound the error between target values and the true values on the critical set.

**Lemma A.2.** *Assume that every leaf-edge value contributing to $\hat{Q}(s,a)$ for any $(s,a) \in \mathcal{C}$ is estimated by at least $n_{\min}^{\text{tr}}$ i.i.d. terminal rollouts under $\pi_u$, each bounded in $[0,1]$. Then with probability at least $1 - \delta$,*

$$\sup_{(s,a) \in \mathcal{C}} \left| \hat{Q}(s,a) - Q^{\pi_u}(s,a) \right| \leq \epsilon_{\text{tr}}(\delta) := \sqrt{\frac{\ln(2|\mathcal{C}|/\delta)}{2n_{\min}^{\text{tr}}}}$$

*Proof.* Fix $(s,a) \in \mathcal{C}$. Let $Z_1, Z_2, \dots, Z_n$ be the i.i.d. terminal rollout returns contributing to $\hat{Q}$, where $n \geq n_{\min}^{\text{tr}}$. Each $Z_i \in [0,1]$ and by (17), $\mathbb{E}[Z_i] = Q^{\pi_u}(s,a)$.

The target estimate is $\hat{Q} = \frac{1}{n} \sum_{i=1}^{n} Z_i$.

By Hoeffding's inequality for bounded random variables:

$$\Pr\left( \left| \hat{Q} - Q^{\pi_u}(s,a) \right| \geq \epsilon \right) \leq 2 \exp\left( -2n\epsilon^2 \right) \leq 2 \exp\left( -2n_{\min}^{\text{tr}}\epsilon^2 \right). \tag{26}$$

Setting $\epsilon = \sqrt{\frac{\ln(2|\mathcal{C}|/\delta)}{2n_{\min}^{\text{tr}}}}$, we obtain

$$\Pr\left( \left| \hat{Q} - Q^{\pi_u}(s,a) \right| \geq \epsilon \right) \leq \frac{\delta}{|\mathcal{C}|}. \tag{27}$$

Taking a union bound over all $|\mathcal{C}|$ pairs in $\mathcal{C}$:

$$\Pr\left( \sup_{(s,a) \in \mathcal{C}} \left| \hat{Q}(s,a) - Q^{\pi_u}(s,a) \right| \geq \epsilon \right) \leq |\mathcal{C}| \cdot \frac{\delta}{|\mathcal{C}|} = \delta. \tag{28}$$

Thus with probability at least $1 - \delta$, the stated bound holds. $\qquad \square$

Lemma A.2 shows that the target error bound is tightened as the number of samples increases.

**Learning Error.** We now relate the learning error of the Q-model $Q_\theta$ to the training sample size. Let $S = \{(x_i, y_i)\}_{i=1}^{m}$ be the training data, where $x_i = (s_i, a_i)$ denotes a state-action pair and $y_i = \hat{Q}(s_i, a_i) \in [0,1]$ is the corresponding target value. We analyze the binary cross-entropy loss:

$$\ell(p, y) = -y \log p - (1-y) \log(1-p) \tag{29}$$

For analysis, we restrict predictors to $[\tau, 1 - \tau]$ for some $\tau \in (0, 1/2)$, solely to ensure $\ell(\cdot, y)$ is Lipschitz with a finite constant $L := 1/\tau$.

Define the expected and empirical risks under $\mathcal{D}$:

$$\mathcal{R}(Q) := \mathbb{E}_{(X,Y)\sim\mathcal{D}}[\ell(Q(X), Y)], \tag{30}$$

$$\hat{\mathcal{R}}_S(Q) := \frac{1}{m}\sum_{i=1}^{m}\ell(Q(x_i), y_i).$$

Let $\mathcal{Q}$ be the function class and $\hat{\mathfrak{R}}_m(\mathcal{Q})$ the empirical Rademacher complexity on $\{x_i\}_{i=1}^m$.

**Lemma A.3** (Rademacher generalization bound). *With probability at least $1 - \delta$ over the draw of $S$,*

$$\sup_{Q\in\mathcal{Q}}\left|\mathcal{R}(Q) - \hat{\mathcal{R}}_S(Q)\right| \le 2L\,\hat{\mathfrak{R}}_m(\mathcal{Q}) + 3\sqrt{\frac{\ln(2/\delta)}{2m}}. \tag{31}$$

*Proof.* The proof proceeds in four steps.

**Step 1: Symmetrization.** Let $S = \{(x_i, y_i)\}_{i=1}^m$ and $S' = \{(x_i', y_i')\}_{i=1}^m$ be two independent samples from $\mathcal{D}$. By standard symmetrization arguments (see, e.g., Theorem 26.5 in Shalev-Shwartz & Ben-David, 2014):

$$\mathbb{E}_S\left[\sup_{Q\in\mathcal{Q}}\left(\mathcal{R}(Q) - \hat{\mathcal{R}}_S(Q)\right)\right] \le 2\mathbb{E}_{S,\sigma}\left[\sup_{Q\in\mathcal{Q}}\frac{1}{m}\sum_{i=1}^{m}\sigma_i\ell(Q(x_i), y_i)\right], \tag{32}$$

where $\sigma_1, \ldots, \sigma_m$ are i.i.d. Rademacher random variables ($\Pr(\sigma_i = \pm 1) = 1/2$).

**Step 2: Lipschitz contraction.** Since $\ell(\cdot, y)$ is $L$-Lipschitz on $[\tau, 1-\tau]$ (with $L = 1/\tau$), and the Rademacher complexity satisfies the contraction principle:

$$\mathbb{E}_{S,\sigma}\left[\sup_{Q\in\mathcal{Q}}\frac{1}{m}\sum_{i=1}^{m}\sigma_i\ell(Q(x_i), y_i)\right] \le L \cdot \mathbb{E}_{S,\sigma}\left[\sup_{Q\in\mathcal{Q}}\frac{1}{m}\sum_{i=1}^{m}\sigma_i Q(x_i)\right] = L\hat{\mathfrak{R}}_m(\mathcal{Q}). \tag{33}$$

**Step 3: McDiarmid's inequality.** Define $\Phi(S) := \sup_{Q\in\mathcal{Q}}\left(\mathcal{R}(Q) - \hat{\mathcal{R}}_S(Q)\right)$. Changing a single sample $(x_i, y_i)$ changes $\Phi(S)$ by at most $\frac{2\|\ell\|_\infty}{m} \le \frac{2}{m}$ (since losses are bounded when outputs are in $[\tau, 1-\tau]$ and labels in $[0,1]$). By McDiarmid's inequality:

$$\Pr\left(\Phi(S) - \mathbb{E}[\Phi(S)] \ge t\right) \le \exp\left(-\frac{2t^2}{m\cdot(2/m)^2}\right) = \exp\left(-\frac{mt^2}{2}\right). \tag{34}$$

Setting $t = \sqrt{\frac{\ln(2/\delta)}{2m}}$ and combining:

$$\Pr\left(\sup_{Q\in\mathcal{Q}}\left(\mathcal{R}(Q) - \hat{\mathcal{R}}_S(Q)\right) \ge 2L\hat{\mathfrak{R}}_m(\mathcal{Q}) + \sqrt{\frac{\ln(2/\delta)}{2m}}\right) \le \frac{\delta}{2}. \tag{35}$$

**Step 4: Two-sided bound.** Applying the same argument to $\hat{\mathcal{R}}_S(Q) - \mathcal{R}(Q)$ and taking a union bound yields the two-sided result with probability $1 - \delta$. The constant 3 (instead of 2) in the final bound accounts for technical refinements in the symmetrization step. $\square$

We now convert the uniform bound into an excess-risk bound between the learned predictor and the best achievable predictor. we assume approximate ERM:

$$\hat{\mathcal{R}}_S(Q_\theta) \leq \inf_{Q \in \mathcal{Q}} \hat{\mathcal{R}}_S(Q) + \epsilon_{\text{opt}}, \tag{36}$$

where $\epsilon_{\text{opt}} \geq 0$ is the optimization error. Define the in-class optimal predictor

$$Q_{\mathcal{Q}}^\star := \arg\min_{Q \in \mathcal{Q}} \mathcal{R}(Q). \tag{37}$$

Then we can have the excess risk bound.

**Lemma A.4** (Excess risk bound)**.** *On the event of Lemma A.3, we have*

$$\mathcal{R}(Q_\theta) - \mathcal{R}(Q_{\mathcal{Q}}^\star) \leq 2\varepsilon_{\text{gen}}(m, \delta) + \epsilon_{\text{opt}}, \tag{38}$$

*where*

$$\varepsilon_{\text{gen}}(m, \delta) := 2L\,\hat{\mathfrak{R}}_m(\mathcal{Q}) + 3\sqrt{\frac{\ln(2/\delta)}{2m}}. \tag{39}$$

*Proof.* On the event of Lemma A.3, for all $Q \in \mathcal{Q}$:

$$\mathcal{R}(Q) \leq \hat{\mathcal{R}}_S(Q) + \varepsilon_{\text{gen}}, \tag{40}$$
$$\hat{\mathcal{R}}_S(Q) \leq \mathcal{R}(Q) + \varepsilon_{\text{gen}}. \tag{41}$$

Now we bound the excess risk:

$$\begin{aligned}
\mathcal{R}(Q_\theta) - \mathcal{R}(Q_{\mathcal{Q}}^\star) &\leq \hat{\mathcal{R}}_S(Q_\theta) + \varepsilon_{\text{gen}} - \mathcal{R}(Q_{\mathcal{Q}}^\star) && \text{(by (40) applied to } Q_\theta) & (42)\\
&\leq \inf_{Q \in \mathcal{Q}} \hat{\mathcal{R}}_S(Q) + \epsilon_{\text{opt}} + \varepsilon_{\text{gen}} - \mathcal{R}(Q_{\mathcal{Q}}^\star) && \text{(by approximate ERM (36))} & (43)\\
&\leq \hat{\mathcal{R}}_S(Q_{\mathcal{Q}}^\star) + \epsilon_{\text{opt}} + \varepsilon_{\text{gen}} - \mathcal{R}(Q_{\mathcal{Q}}^\star) && \text{(since } Q_{\mathcal{Q}}^\star \in \mathcal{Q}) & (44)\\
&\leq \mathcal{R}(Q_{\mathcal{Q}}^\star) + \varepsilon_{\text{gen}} + \epsilon_{\text{opt}} + \varepsilon_{\text{gen}} - \mathcal{R}(Q_{\mathcal{Q}}^\star) && \text{(by (41) applied to } Q_{\mathcal{Q}}^\star) & (45)\\
&= 2\varepsilon_{\text{gen}} + \epsilon_{\text{opt}}. && & (46)
\end{aligned}$$

$\square$

In our context, the loss function $\ell(p, y)$ is a strictly proper scoring rule for Bernoulli distributions. Thus, we can use Pinsker's inequality to obtain the following direct link from risk to $L_2$ error. Let $Q^\dagger(x) := \mathbb{E}[Y \mid X = x]$ denote the true conditional expectation under $\mathcal{D}$. In our setting, because training targets are generated from terminal rollouts under $\pi_u$ and unbiased Bellman backups (Lemma A.1), we have $Q^\dagger(x) = Q^{\pi_u}(x)$.

**Lemma A.5.** *Let $Q^\dagger(x) := \mathbb{E}[Y \mid X = x]$ denote the true conditional expectation under $\mathcal{D}$. Then for any predictor $Q$,*

$$\mathbb{E}_{X \sim \mathcal{D}}\left[\left(Q(X) - Q^\dagger(X)\right)^2\right] \leq \frac{1}{2}\left(\mathcal{R}(Q) - \mathcal{R}\left(Q^\dagger\right)\right).$$

*Proof.* The proof proceeds in three steps: (i) express excess log-loss as KL divergence, (ii) apply Pinsker's inequality, (iii) specialize to Bernoulli distributions.

**Step 1: Log-loss decomposition.** For the log-loss $\ell(p, y) = -y \log p - (1-y) \log(1-p)$ with $p, y \in (0, 1)$, we have the identity:

$$\ell(p, y) = \ell(y, y) + \text{KL}(\text{Bern}(y)\|\text{Bern}(p)), \tag{47}$$

where $\text{KL}(\text{Bern}(y)\|\text{Bern}(p)) = y \log \frac{y}{p} + (1-y) \log \frac{1-y}{1-p}$.

*Verification:*

$$\ell(y, y) + \mathrm{KL}(\mathrm{Bern}(y)\|\mathrm{Bern}(p)) \tag{48}$$

$$= [-y \log y - (1-y)\log(1-y)] + \left[y \log \frac{y}{p} + (1-y) \log \frac{1-y}{1-p}\right] \tag{49}$$

$$= -y \log y - (1-y)\log(1-y) + y \log y - y \log p + (1-y)\log(1-y) - (1-y)\log(1-p) \tag{50}$$

$$= -y \log p - (1-y)\log(1-p) \tag{51}$$

$$= \ell(p, y). \tag{52}$$

**Step 2: Pinsker's inequality.** The classical Pinsker's inequality states that for any two probability distributions $P$ and $Q$:

$$\mathrm{TV}(P, Q)^2 \leq \frac{1}{2}\mathrm{KL}(P\|Q), \tag{53}$$

where $\mathrm{TV}(P, Q) = \sup_A |P(A) - Q(A)|$ is the total variation distance.

**Step 3: Specialization to Bernoulli.** For Bernoulli distributions $\mathrm{Bern}(y)$ and $\mathrm{Bern}(p)$:

$$\mathrm{TV}(\mathrm{Bern}(y), \mathrm{Bern}(p)) = |y - p|. \tag{54}$$

(This follows by taking $A = \{1\}$ in the TV definition.)

Applying Pinsker's inequality:

$$(y - p)^2 \leq \frac{1}{2}\mathrm{KL}(\mathrm{Bern}(y)\|\mathrm{Bern}(p)) = \frac{1}{2}\left(\ell(p, y) - \ell(y, y)\right). \tag{55}$$

**Step 4: Conditional expectation and integration.** Fix $x$ and let $y = Q^\dagger(x) = \mathbb{E}[Y \mid X = x]$. Taking the prediction $p = Q(x)$:

$$(Q(x) - Q^\dagger(x))^2 \leq \frac{1}{2}\left(\ell(Q(x), Q^\dagger(x)) - \ell(Q^\dagger(x), Q^\dagger(x))\right). \tag{56}$$

Taking expectation over $X \sim \mathcal{D}$:

$$\mathbb{E}_X\left[(Q(X) - Q^\dagger(X))^2\right] \leq \frac{1}{2}\mathbb{E}_X\left[\ell(Q(X), Q^\dagger(X)) - \ell(Q^\dagger(X), Q^\dagger(X))\right] \tag{57}$$

$$= \frac{1}{2}\left(\mathbb{E}_X[\ell(Q(X), Q^\dagger(X))] - \mathbb{E}_X[\ell(Q^\dagger(X), Q^\dagger(X))]\right). \tag{58}$$

Since $Q^\dagger(x) = \mathbb{E}[Y \mid X = x]$ minimizes the conditional expected log-loss, we have

$$\mathbb{E}_{Y|X}[\ell(Q(X), Y)] \geq \mathbb{E}_{Y|X}[\ell(Q^\dagger(X), Y)] = \ell(Q^\dagger(X), Q^\dagger(X)) + H(Y|X), \tag{59}$$

where $H(Y|X)$ is the conditional entropy (which cancels in the difference).

Thus:

$$\mathbb{E}_X\left[(Q(X) - Q^\dagger(X))^2\right] \leq \frac{1}{2}\left(\mathcal{R}(Q) - \mathcal{R}(Q^\dagger)\right). \tag{60}$$

$\square$

We now prove theorem 5.2.

**Theorem A.6** (Restatement of Theorem 5.2). *With probability at least $1 - \delta$, the learned predictor $Q_\theta$ satisfies*

$$\|Q_\theta - Q^{\pi_u}\|_{2,\rho_\mathcal{C}} \leq \epsilon_{\text{bias}}(m, \delta), \tag{61}$$

*where*

$$\epsilon_{\text{bias}}(m, \delta) := \sqrt{\frac{1}{2}\left(\epsilon_{\text{approx}} + 2\varepsilon_{\text{gen}}(m, \delta/2) + \epsilon_{\text{opt}}\right)}, \tag{62}$$

*with $\epsilon_{\text{approx}} := \mathcal{R}(Q_\mathcal{Q}^\star) - \mathcal{R}(Q^\dagger) \geq 0$.*

*Proof.* The proof proceeds in five steps.

**Step 1: Decompose total excess risk.** We decompose the excess risk of $Q_\theta$ relative to $Q^\dagger$:

$$\mathcal{R}(Q_\theta) - \mathcal{R}(Q^\dagger) = \underbrace{\left(\mathcal{R}(Q_\theta) - \mathcal{R}(Q_\mathcal{Q}^\star)\right)}_{\text{estimation + optimization}} + \underbrace{\left(\mathcal{R}(Q_\mathcal{Q}^\star) - \mathcal{R}(Q^\dagger)\right)}_{\text{approximation}}. \tag{63}$$

**Step 2: Bound estimation + optimization error.** By Lemma A.4 with confidence $\delta/2$:

$$\mathcal{R}(Q_\theta) - \mathcal{R}(Q_\mathcal{Q}^\star) \leq 2\varepsilon_{\text{gen}}(m, \delta/2) + \epsilon_{\text{opt}}. \tag{64}$$

**Step 3: Identify approximation error.** The approximation error is:

$$\epsilon_{\text{approx}} := \mathcal{R}(Q_\mathcal{Q}^\star) - \mathcal{R}(Q^\dagger) \geq 0, \tag{65}$$

which is non-negative since $Q^\dagger$ is the Bayes-optimal predictor (minimizer of population risk over all measurable functions).

**Step 4: Combine and apply Lemma A.5.** With probability at least $1 - \delta/2$:

$$\mathcal{R}(Q_\theta) - \mathcal{R}(Q^\dagger) \leq \epsilon_{\text{approx}} + 2\varepsilon_{\text{gen}}(m, \delta/2) + \epsilon_{\text{opt}}. \tag{66}$$

By Lemma A.5:

$$\mathbb{E}_{X \sim \mathcal{D}}\left[(Q_\theta(X) - Q^\dagger(X))^2\right] \leq \frac{1}{2}\left(\mathcal{R}(Q_\theta) - \mathcal{R}(Q^\dagger)\right). \tag{67}$$

**Step 5: Identify $Q^\dagger = Q^{\pi_u}$ and conclude.** In our setting, training targets are generated from unbiased terminal rollouts under $\pi_u$ with Bellman backups (Lemma A.1). Thus $Q^\dagger(s, a) = \mathbb{E}[Y \mid X = (s, a)] = Q^{\pi_u}(s, a)$.

Taking square roots:

$$\|Q_\theta - Q^{\pi_u}\|_{2,\rho_\mathcal{C}} = \sqrt{\mathbb{E}_{(S,A) \sim \rho_\mathcal{C}}\left[(Q_\theta(S, A) - Q^{\pi_u}(S, A))^2\right]} \tag{68}$$

$$\leq \sqrt{\frac{1}{2}\left(\epsilon_{\text{approx}} + 2\varepsilon_{\text{gen}}(m, \delta/2) + \epsilon_{\text{opt}}\right)} \tag{69}$$

$$= \epsilon_{\text{bias}}(m, \delta). \tag{70}$$

$\square$

### A.2. Proof of Theorem 5.3

**Theorem A.7** (Restatement of Theorem 5.3). *Suppose the bias consistency condition holds: $\epsilon_{\text{bias}}(m, \delta) < \Delta_{\min}^*/2$. Let $\Delta_{\text{eff}} := \Delta_{\min}^* - 2\epsilon_{\text{bias}}(m, \delta) > 0$. Then for the greedy path $\hat{a}_t = \arg\max_{a \in \mathcal{A}(s_t^*)} \bar{Q}_n(s_t^*, a)$ to coincide with the optimal path with probability at least $1 - \delta$, it suffices that*

$$n \geq \frac{32(K-1)c^2 \ln(Hn/\delta)}{\Delta_{\text{eff}}^2} + 2(K-1)\left(2N_0 + \frac{\pi^2}{3}\right). \tag{71}$$

*Proof.* The proof adapts the UCT analysis of Kocsis & Szepesvári (2006) to our guided-MCTS setting.

**Step 1: Setup and notation.** At each decision node $s_t^*$, UCT treats action selection as a multi-armed bandit with $K = |\mathcal{A}(s_t^*)|$ arms. Let $T_a(n)$ denote the number of times action $a$ is selected after $n$ total simulations. The payoff distributions are non-stationary because subtree estimates evolve with exploration.

**Step 2: Apply Kocsis-Szepesvári Theorem 1.** By Theorem 1 of (Kocsis & Szepesvári, 2006), for UCB1 applied to a non-stationary bandit with bias (drift) bounded by $\epsilon$, each suboptimal arm $a$ with gap $\Delta_a > 2\epsilon$ satisfies:

$$\mathbb{E}[T_a(n)] \leq \frac{16c^2 \ln n}{(\Delta_a - 2\epsilon)^2} + 2N_0 + \frac{\pi^2}{3}. \tag{72}$$

In our setting, the bias is $\epsilon = \epsilon_{\text{bias}}(m, \delta)$ (from Theorem 5.2), and for each suboptimal action $a \neq a_t^*$:

$$\Delta_a := Q^{\pi_u}(s_t^*, a_t^*) - Q^{\pi_u}(s_t^*, a) \geq \Delta_{\min}^*. \tag{73}$$

Thus:

$$\mathbb{E}[T_a(n)] \leq \frac{16c^2 \ln n}{(\Delta_a - 2\epsilon_{\text{bias}})^2} + 2N_0 + \frac{\pi^2}{3} \leq \frac{16c^2 \ln n}{\Delta_{\text{eff}}^2} + 2N_0 + \frac{\pi^2}{3}. \tag{74}$$

**Step 3: Sum over suboptimal actions.** Summing over all $K - 1$ suboptimal actions:

$$\sum_{a \neq a_t^*} \mathbb{E}[T_a(n)] \leq \frac{16(K-1)c^2 \ln n}{\Delta_{\text{eff}}^2} + (K-1)\left(2N_0 + \frac{\pi^2}{3}\right). \tag{75}$$

**Step 4: Lower bound optimal action visits.** Since $\sum_a T_a(n) = n$:

$$\mathbb{E}[T_{a_t^*}(n)] = n - \sum_{a \neq a_t^*} \mathbb{E}[T_a(n)] \geq n - \frac{16(K-1)c^2 \ln n}{\Delta_{\text{eff}}^2} - (K-1)\left(2N_0 + \frac{\pi^2}{3}\right). \tag{76}$$

**Step 5: Condition for optimal action dominance.** For the optimal action to be selected (i.e., have the highest empirical mean), it suffices that $\mathbb{E}[T_{a_t^*}(n)] > n/2$, which requires:

$$n > \frac{32(K-1)c^2 \ln n}{\Delta_{\text{eff}}^2} + 2(K-1)\left(2N_0 + \frac{\pi^2}{3}\right). \tag{77}$$

**Step 6: Union bound over path.** The optimal path has $H$ decision points. Taking a union bound:

$$\Pr(\text{all } \hat{a}_t = a_t^*) \geq 1 - H \cdot \frac{\delta}{2H} = 1 - \frac{\delta}{2}. \tag{78}$$

Combined with the $1 - \delta/2$ probability from the bias consistency guarantee (Theorem 5.2), the total success probability is at least $1 - \delta$. $\qquad\square$

# B. Additional Experimental Results

This appendix provides detailed analyses of the additional experimental results presented in the supplementary figures. We systematically conduct cross-platform generalization, training dynamics, hyperparameter sensitivity, and other experimental analyses for the proposed Executable Agentic Memory (EAM) framework.

## B.1. Additional Results on OSWorld

We additionally evaluate EAM on OSWorld to examine cross-platform generalization beyond mobile benchmarks. As shown in Table 4, EAM achieves a 51.8% success rate, substantially outperforming standalone cloud models such as GPT-5.2 (47.3%) and Claude-Sonnet-4.5 (42.9%), as well as on-device baselines. EAM also remains competitive with GUI-Explorer (51.8% vs. 52.6%) while consuming far fewer tokens per step (8.9K vs. 70.2K). These results suggest that executable memory provides effective cross-platform generalization while preserving the token-efficiency advantage of our plan-then-execute framework.

| Method | Type | Base Model | Success Rate (%) | Tokens/Step (K) |
|---|---|---|---|---|
| GPT-5.2 | Cloud | GPT-5.2 | 47.3 | 48.3 |
| Claude-Sonnet-4.5 | Cloud | Claude-Sonnet-4.5 | 42.9 | 47.7 |
| UI-TARS-1.5-7B | On-device | UI-TARS-1.5-7B | 27.3 | - |
| UI-TARS-2B | On-device | UI-TARS-2B | 3.1 | - |
| AppAgentX | Framework | GPT-5.2 | 49.3 | 12.3 |
| GUI-Explorer | Framework | GPT-5.2 | 52.6 | 70.2 |
| EAM (Ours) | Framework | GPT-5.2, Qwen2.5-3B-ft, UI-TARS-2B | **51.8** | **8.9** |

*Table 4.* Performance on OSWorld. "Tokens/Step" denotes the average API token consumption per interaction step. "-" indicates locally deployed models without API calls.

## B.2. Training Loss Dynamics Across Self-Learning Rounds

Fig. 6 presents the training loss curves across four self-learning rounds for three model sizes (Qwen2.5-0.5B-Instruct, Qwen2.5-1.5B-Instruct, and Qwen2.5-3B-Instruct) evaluated on three benchmarks (AndroidWorld, DroidTask, and MobileMiniWob).

**Progressive Loss Reduction.** A salient pattern emerges across all configurations: the initial loss at the beginning of each subsequent round starts consistently lower than the previous round. This progressive reduction in starting loss demonstrates that the Q-model successfully retains and builds upon knowledge acquired in previous iterations, validating the effectiveness of our iterative self-training pipeline. Formally, this observation suggests that the empirical risk $\hat{\mathcal{R}}_{\mathcal{S}}(Q_\theta)$ decreases across rounds, which according to Lemma A.4, implies corresponding reductions in the true risk $\mathcal{R}(Q_\theta)$.

**Convergence Characteristics.** All training curves exhibit rapid initial descent followed by stabilization, typically converging within the first 100–150 training steps. The converged loss values decrease monotonically across rounds, indicating that the quality of self-generated training data improves as the Q-model becomes more accurate at value estimation. This phenomenon creates a virtuous cycle: better value estimates lead to more informative MCTS rollouts, which in turn produce higher-quality Bellman backup targets, further improving subsequent training rounds.

**Model Capacity Effects.** Larger models consistently achieve lower final loss values across all benchmarks. This performance gap reflects the increased representational capacity of larger models to capture complex relationships between GUI states, actions, and their associated Q-values. The relationship can be understood through the lens of approximation error $\epsilon_{\text{approx}}$ in Theorem 5.2: larger model classes $\mathcal{Q}$ reduce the gap $\mathcal{R}(Q_{\mathcal{Q}}^\star) - \mathcal{R}(Q^\dagger)$, yielding tighter bias bounds.

## B.3. Effect of MCTS Iteration Count

Figure 7 investigates the trade-off between MCTS simulation budget and performance by varying the number of iterations $N \in \{10, 30, 50, 100\}$. We measure both success rate and path extraction latency across all three benchmarks using the 3B model.

**Monotonic Performance Scaling.** Success rate increases monotonically with MCTS iterations across all benchmarks that additional simulation budget enables more thorough exploration of the search space, increasing the probability of discovering optimal paths.

**Diminishing Returns.** The performance gains exhibit pronounced diminishing returns. Quantitatively, the marginal improvement per additional 10 iterations decreases substantially:

- $N : 10 \rightarrow 30$: +10 points on AndroidWorld

- $N : 30 \rightarrow 50$: +6 points on AndroidWorld

- $N : 50 \rightarrow 100$: 0.7 points on AndroidWorld

This sublinear scaling suggests that moderate iteration counts capture most of the benefit from tree search, with additional simulations primarily refining already-promising paths rather than discovering qualitatively better alternatives.

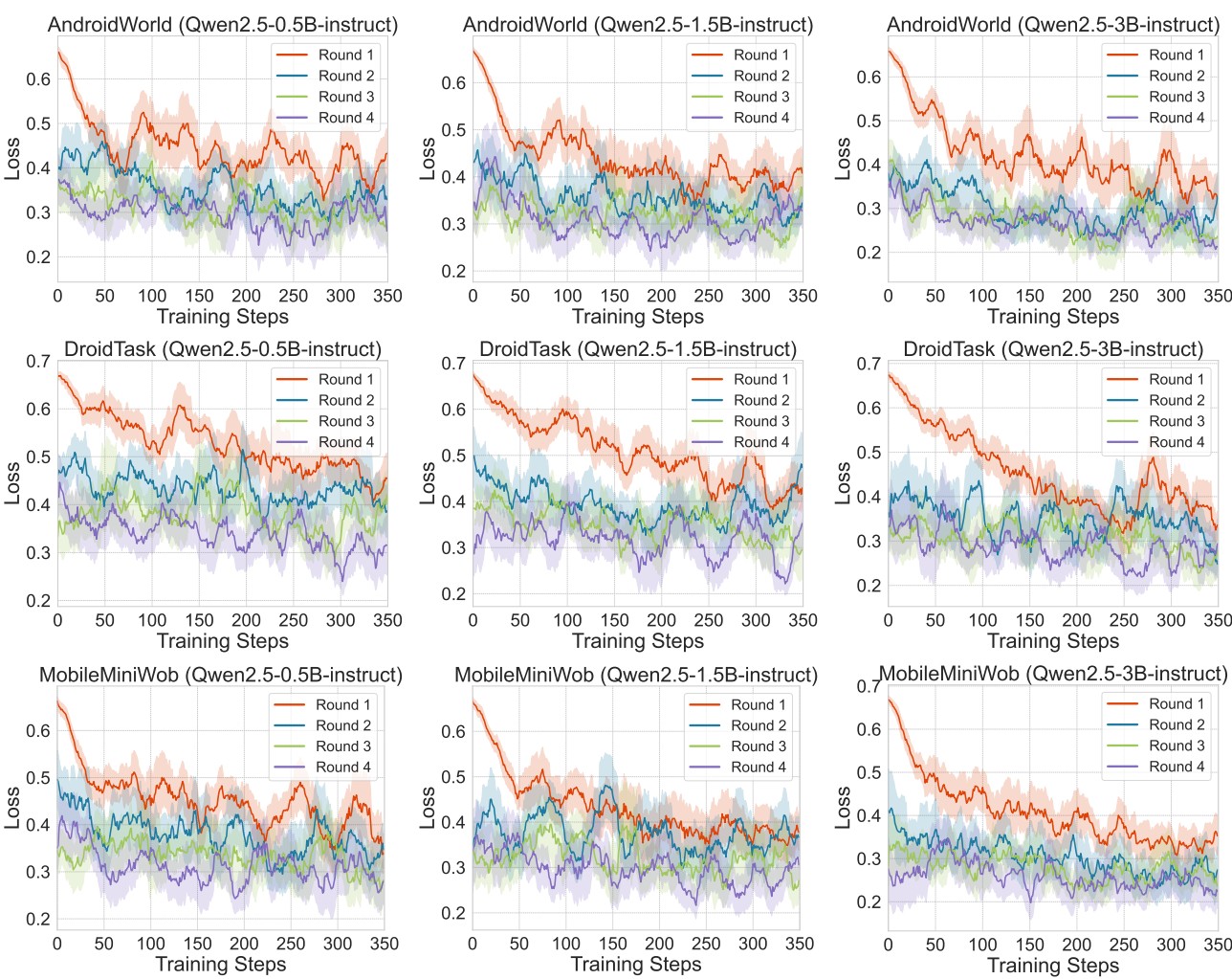

*Figure 6.* Training loss curves across self-learning rounds. We show the training loss for each round across different model sizes and benchmarks.

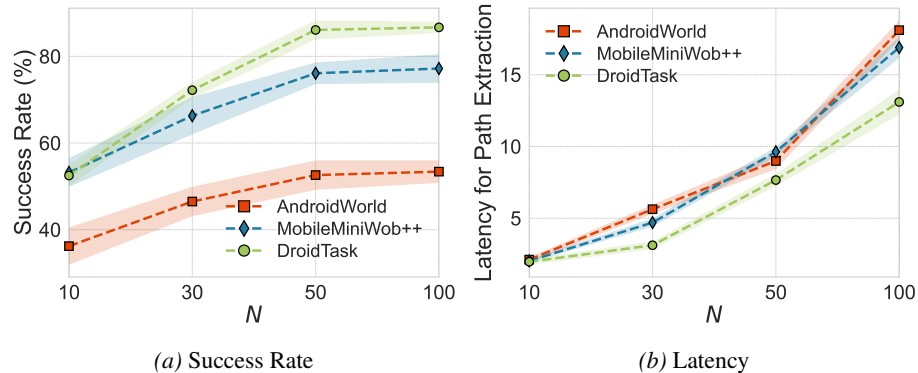

*(a)* Success Rate            *(b)* Latency

*Figure 7.* The impact of different number of MCTS iterations on performance.

**Latency Characteristics.** Path extraction latency scales approximately linearly with iteration count. This linear scaling, combined with the sublinear performance gains, implies a favorable efficiency trade-off at moderate iteration counts.

**Practical Operating Points.** The results suggest $N = 30$ or $N = 50$ as practical operating points. At $N = 50$, the system achieves over $95\%$ of the $N = 100$ performance on all benchmarks while requiring only $50\%$ of the computation time.

**Connection to Theoretical Bounds.** These findings align with Theorem 5.3, which establishes simulation complexity scaling as:

$$n \geq \frac{32(K-1)c^2 \ln(Hn/\delta)}{\Delta_{\text{eff}}^2} + 2(K-1)\left(2N_0 + \frac{\pi^2}{3}\right). \tag{79}$$

The logarithmic dependence on $n$ in the bound is consistent with the observed diminishing returns. The empirical observation that moderate $N$ suffices suggests that practical task instances have relatively large effective action gaps $\Delta_{\text{eff}} = \Delta_{\text{min}}^* - 2\epsilon_{\text{bias}}$, enabling efficient path recovery without exhaustive search.

### B.4. Performance with Stronger Models

| Method | Success Rate (%) |
|---|---|
| GPT-5.2 | 56.9 |
| Claude-Sonnet-4.5 | 54.3 |
| UI-TARS-7B-DPO | 30.1 |
| GUI-Owl-7B | 44.0 |
| EAM (GPT-5.2 + UI-TARS-7B-DPO) | 61.2 |
| EAM (GPT-5.2 + GUI-Owl-7B) | **66.4** |

*Table 5.* AndroidWorld results with stronger models. The first four rows report standalone model performance, while the last two rows integrate stronger planning and execution models into EAM.

The main AndroidWorld result in the paper uses UI-TARS-2B as the on-device execution model, prioritizing deployment efficiency. Since EAM is designed as a plug-and-play framework, we further evaluate whether stronger cloud planners and GUI execution models can be integrated to improve performance. As shown in Table 5, standalone frontier models such as GPT-5.2 and Claude-Sonnet-4.5 achieve 56.9% and 54.3% success rates, while stronger GUI executors such as UI-TARS-7B-DPO and GUI-Owl-7B achieve 30.1% and 44.0%, respectively. When integrated into EAM, performance further improves to 61.2% with UI- TARS-7B-DPO and 66.4% with GUI-Owl-7B. These results indicate that EAM is complementary to stronger base models: better planners and executors can be plugged into the framework, while executable memory and value- guided search still provide additional gains over using the models alone.

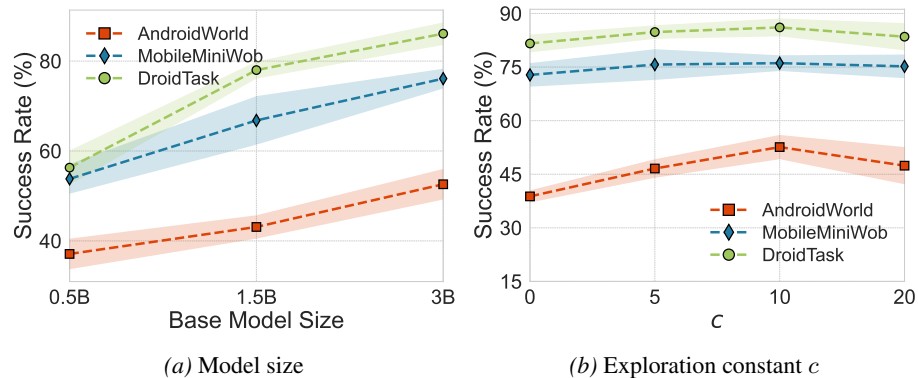

*(a)* Model size        *(b)* Exploration constant $c$

*Figure 8.* The impact of model size and exploration constant on performance.

## B.5. Influence of Model Size and Exploration Constant

Figure 8 presents two hyperparameter studies: (a) the effect of base model size on success rate across 0.5B, 1.5B, and 3B parameters; and (b) the impact of the UCT exploration constant $c \in \{0, 5, 10, 20\}$ on performance, where $c = 0$ corresponds to pure exploitation.

**Model Size Analysis.** Larger models achieve higher success rates across all benchmarks.

**Exploration Constant Analysis.** Performance peaks at $c = 10$ across all benchmarks Both pure exploitation ($c = 0$) and excessive exploration ($c = 20$) yield degraded performance. At $c = 0$, the search degenerates to greedy selection, forfeiting the benefits of exploration demonstrated in Section 6.2. At $c = 20$, the exploration bonus dominates the Q-value term in UCT, causing the search to behave nearly randomly and waste simulation budget on unpromising branches.

**Benchmark-Specific Sensitivity.** AndroidWorld shows the highest sensitivity to $c$. DroidTask and MobileMiniWob++ exhibit more stable performance. This differential sensitivity correlates with task complexity: AndroidWorld's deeper search spaces and longer optimal paths create more opportunities for both beneficial exploration and wasteful over-exploration.

**Theoretical Perspective.** The exploration constant $c$ appears directly in the UCT criterion:

$$\text{UCT}(s, a) = Q(s, a) + c\sqrt{\frac{\ln N(s)}{N(s, a)}}, \tag{80}$$

and in the sample complexity bound of Theorem 5.3, where it contributes to the $c^2$ term in the numerator. The empirical finding that moderate $c$ optimizes performance validates the theoretical exploration-exploitation trade-off: insufficient exploration risks committing to suboptimal paths before adequately evaluating alternatives, while excessive exploration incurs unnecessary simulation cost.

## B.6. Effect of Offline Exploration Coverage

| Exploration Ratio | AndroidWorld (%) | MobileMiniWob++ (%) |
|---|---|---|
| 0% (no KG) | 6.9 | 31.5 |
| 75% | 42.2 | 66.7 |
| 100% (full EAM) | **52.6** | **76.1** |

*Table 6.* Success rate under different offline exploration ratios. The 0% setting corresponds to no KG guidance, while 100% denotes the full EAM construction.

To quantify how offline trace coverage affects downstream performance, we vary the proportion of explored tasks used to construct the KG. As shown in Table 6, performance improves consistently as exploration coverage increases. Without KG guidance, the success rate is only 6.9% on AndroidWorld and 31.5% on MobileMiniWob++. With 75% exploration coverage, EAM already reaches 42.2% and 66.7%, demonstrating that partial offline memory provides substantial gains.

Full exploration further improves performance to 52.6% and 76.1%, confirming that broader task-oriented traces lead to a more complete executable memory and more reliable path extraction.

### B.7. Effect of State Checker in Offline Exploration

| State Checker Model | Success Rate (%) |
|---|---|
| Qwen2.5-VL-72B-Instruct | 49.1 |
| GPT-4o | 55.2 |
| GPT-5.2 | **59.5** |

*Table 7.* Impact of the state checker model on AndroidWorld. Sub-goal generation and task execution are fixed as GPT-5.2, and only the state checker model is varied.

Since intermediate exploration states do not have ground-truth labels, directly measuring state checker accuracy is difficult. We therefore evaluate its effect indirectly by varying only the state checker model while keeping sub-goal generation and task execution fixed as GPT-5.2. As shown in Table 7, stronger state checkers lead to better downstream performance: replacing Qwen2.5-VL-72B-Instruct with GPT-4o improves the success rate from 49.1% to 55.2%, and GPT-5.2 further improves it to 59.5%. These results suggest that state checking is a critical source of offline memory quality, because incorrect continuation or termination decisions can either introduce noisy branches into the KG or miss valid execution paths.

## C. Training Data

We train our Q-model in two phases, including initialization training and self-training iterations.

### C.1. Step-level Preference Dataset for Initialization Training

Our training data pairs are derived from AMEX, which contains a large collection of expert trajectories with semantic descriptions for each action step, as well as semantic annotations for task-irrelevant elements on each page. For each page in the expert trajectories, we construct preference optimization pairs consisting of (expert action description, multiple task-irrelevant element descriptions) as follows:

```
{
  "instruction": "Open Google_Tasks.  Delete all completed tasks in the \"Work
List\".",
  "history_actions": [],
  "page_caption": "The page displays a list of tasks organized by categories such as
work, health, and family, with options to mark tasks as complete, add stars, and view
details.",
  "correct_actions": [
    "View the list of completed tasks"
  ],
  "false_actions": [
    "View details of the task 'submit progress report' due on Monday, April 15",
    "View details of the task 'project x' with 1 subtask",
    "View tasks under 'work' category",
    "View details of the task 'send a draft to the team' due on Friday, April 12"
  ]
}
```

### C.2. Dataset Generated by MCTS Rollouts

Following initialization, the value model is iteratively trained to predict the Q-value of state-action pairs. An example of dataset from AndroidWorld is illustrated below.

**Input:**
```
<|user|>:
Task:  Add the expenses from expenses.jpg in Simple Gallery Pro to pro expense.
You are at:  This is an expense tracking dashboard that allows users to monitor their
spending patterns through a weekly calendar view and detailed transaction history,
while providing quick access to add new expenses via the floating action button.
Executed path:  Start of Task
Proposed action:
```

**Output:**
```
<|assistant|>:  Navigate from expense tracking application to gallery app to locate and
review expense-related images or receipts for reference during expense entry
workflow<end_of_step>
```

**Label:**  0.7679166666666666

## D. Prompts

### D.1. Prompt for Sub-goals Generation at Offline Exploration Phase

```
Given a user task, the current screenshot of {app_name}, and available UI elements,
generate multiple potential sub-goals to progress toward completing the user task.
Each sub-goal must:

1.  Start with interacting with a specific UI element from the provided element list
2.  Be expressed as a single, clear directive following the pattern:  [Starting action]
+ [Specific steps] + [End goal]
3.  Be achievable within approximately 3 actions (AT MOST 5) from the anchor element
4.  Provide a concrete target state that advances toward the user task completion

 Context Information:
- User Task:  {user_task}
- App name:  {app_name}
- Package name:  {package_name}
- Current screen elements (Only interact with *visible=true elements):
  {element_list}
- Activity context:  {activity_list}
- Recent History Action (up to 5):  {action_history}
- Sub-goals History:  {subgoal_history}
- State Summary:  {state_summary}

Task Execution Analysis:
Before generating new sub-goals, analyze the current execution state:

1.  Completed Progress:  Review the sub-goals history to understand what has already
been accomplished toward the main task
2.  Current Position:  Based on the state summary and recent actions, identify where
you are in the task workflow
3.  Remaining Work:  Determine what specific components of the user task still need to
be completed
4.  Next Logical Steps:  Identify the most logical next actions that build upon
completed sub-goals

Sub-goal Generation Strategy:
- Continuation Focus:  Generate sub-goals that logically continue from where previous
sub-goals left off
- Avoid Redundancy:  Do not repeat actions or objectives that have already been
successfully completed
- Progressive Advancement:  Each sub-goal should represent a clear step forward in the
overall task completion
```

```
For each sub-goal, provide:
1.  Anchor Element:  The specific UI element ID/description from the list to start with
2.  Sub-goal:  Single directive sentence following [Starting action] + [Specific steps]
+ [End goal] pattern
3.  Confidence Score:  How likely this sub-goal is to advance toward task completion
(0.0-1.0)

Format each sub-goal as:
Sub-goal [N]:
Anchor:  [Element ID/description from element_list]
Directive:  [Single clear instruction with starting action + steps + end goal]
Confidence:  [0.0-1.0]
```

## D.2. Prompt for Progress Evaluation during Exploration

```
Given the user task, action history, and current screenshot of {app_name}, evaluate the
current exploration state and determine the next action strategy.

Context Information:
- User Task:  {user_task}
- App name:  {app_name}
- Package name:  {package_name}
- Recent Action History:  {action_history}
   (if the last action is '{"action_type":  "status", "goal_status":  "complete"}', it
means the last sub-goal was complete successfully)
- Sub-goals History:  {subgoal_history}
- Current screen elements:
   {element_list}

Analysis Requirements:
1.  Compare the current state with the expected end goal of the user task
2.  Evaluate whether the recent actions are leading toward task completion
3.  Assess if the current exploration path is meaningful and relevant
4.  Consider whether all required steps have been executed successfully
5.  Verify if the current screen/state indicates task completion
6.  Account for any error states, dead-ends, or repetitive actions in the history
7.  Make sure to use answer action for information retrieval task
   ({"action_type":  "answer", "text":  "<answer_text>"} is the last action in the action
history)
8.  Be strict about completion – partial progress is not completion

Evaluation Criteria:
- Has the user task's primary objective been achieved?  (COMPLETED)
   * Have we completed all the sub-goals required by the task and at the expected final
state/screen for this task?
- Are we making meaningful progress toward the goal?  (CONTINUE)
- Are we stuck, going in wrong direction, or exploring irrelevant paths?  (BACKTRACK)
- Is there clear evidence of task completion, progress, or deviation in the current
state?

Format your response as:
Reasoning:  [Detailed analysis of current progress, referencing specific elements from
action history, current state, and task relevance.  Explain why we should continue,
backtrack, or if task is complete]
Result:  [CONTINUE/BACKTRACK/COMPLETED]
```

### D.3. Prompt for Action Group Mining

```
You are an AI assistant specialized in generating high-level common UI operation nodes
which can be part of a variety of operations.  You need to generate a complete
description of a high-level action node based on the given chain information.

Please generate a high-level action node based on the following UI operation chain
information:
  Task description:
  {task_description}

  Chain operations:
  {chain_operations}

  Chain element details:
  {element_details}

  Chain reasoning results:
  {reasoning_results}

Please generate a concise description of the high-level action node, including the
following fields:
- action_id:  Generate a unique ID for the high-level action (format like:
"high_level_action_xxx")
- name:  Concise name of the high-level action
- function_description:  Brief description of the action's functionality and purpose
- preconditions:  Required conditions before executing this action, including:
    * task_state:  What task context or state is needed
    * page_state:  What page/interface state must be present
- post_conditions:  Resulting state after completing this action, including:
    * task_state:  How the task context changes
    * page_state:  What page/interface state is reached
- element_sequence:  Simplified sequence of key elements in this action:
    * element_id:  Element ID
    * atomic_action:  Action performed
    * order:  Execution order
```

