# OpenReview forum: "Executable Agentic Memory for GUI Agent"
_ICML.cc/2026/Conference — ICML 2026 regular_

### Official Review · Reviewer_iY2H · 2026-03-12

**Soundness:** 2
**Presentation:** 3
**Significance:** 2
**Originality:** 2
**Overall Recommendation:** 4
**Confidence:** 3

**Summary:**

This paper proposes Executable Agentic Memory (EAM), a framework that improves the robustness of GUI agents by shifting from step-wise LLM reasoning to structured planning over a knowledge graph. The method first performs offline exploration of the GUI environment to construct a knowledge graph of states and actions, and then uses a lightweight Q-model guided Monte Carlo Tree Search (MCTS) to extract executable action paths for a given instruction. Experiments on several GUI benchmarks show improved task success rates compared to multiple on-device baselines while maintaining relatively low inference latency. The results suggest that grounding agent decision-making in structured memory can improve reliability for long-horizon GUI tasks.

**Compliance With Llm Reviewing Policy:**

Affirmed.

**Final Justification:**

The rebuttal addresses most of my concerns, with clearer positioning, additional experiments, and better cost characterization. While I still have some concerns about generality due to the reliance on environment-specific exploration, I find the response overall convincing and am willing to increase my score.

**Key Questions For Authors:**

Please see the weakness part.

**Limitations:**

Yes.

**Strengths And Weaknesses:**

Strength
1. Clear overall framework and motivation.
The paper is well written and easy to follow. The authors provide a clear motivation: previous model-centric and step-wise GUI agents rely on LLM reasoning at every step, which can introduce hallucinations and reduce robustness in long-horizon tasks. This makes the proposed structured memory and planning framework well motivated.
2. Competitive empirical performance.
The proposed method demonstrates strong performance on multiple benchmarks, outperforming several on-device baselines and achieving competitive results compared to stronger cloud-based systems.

Weakness

The proposed framework relies on a pre-exploration phase that constructs a knowledge graph of the GUI environment before the agent begins executing tasks. This design appears misaligned with how humans interact with interfaces: users typically discover relevant elements on demand while pursuing a goal, rather than exhaustively exploring the entire application beforehand. From a technical perspective, this requirement raises concerns about the generality and scalability of the approach, since the system must build and maintain environment-specific knowledge graphs before deployment. The following issues arise from this design:

1. Limited generalization across interfaces.
The knowledge graph, action groups, and transition structures are constructed per environment. As a result, the system appears to require rebuilding the graph when encountering new applications or websites. This limits generalization and scalability, particularly in dynamic environments where GUIs frequently change.
2. Offline cost is not evaluated.
While the paper reports inference latency, the reported efficiency only reflects the online planning stage. The cost of the offline components—including DFS exploration, trajectory collection, knowledge graph construction, action-group mining, and Q-model training—is not quantified. Without measuring these costs (e.g., number of interactions, time required, compute), it is difficult to assess the overall efficiency of the proposed system.
3. Additional training overhead.
The approach introduces a self-training pipeline for the Q-model that requires iterative MCTS search and model updates. The paper does not report the training cost, data requirements, or sensitivity to training rounds, making it unclear how expensive the method is to adapt to new environments.

---

> ### Author Rebuttal · Authors · 2026-03-31
>
> We thank the reviewer for the thoughtful evaluation. The reviewer's main concern centers on the pre-exploration design. We address each sub-point below.
>
> **W1 (Pre-exploration misaligned with human interaction).**
> We respectfully note that the analogy to human on-demand discovery, while intuitive, overlooks a key difference: humans possess vast prior knowledge about GUI conventions accumulated over years of experience, which allows them to interact with new interfaces efficiently. Current GUI agents lack such prior knowledge and must reason from scratch at every step, leading to compounding errors in long-horizon tasks. Our offline exploration serves precisely this role — it equips the agent with structured prior knowledge (the KG) before deployment, analogous to how a human's accumulated experience enables efficient on-demand interaction. Moreover, our exploration is task-oriented rather than exhaustive: real-world users navigate a limited set of repetitive functional paths, and our DFS is guided by task-relevant sub-goals, constraining exploration to the practically reachable portion. These trajectories capture core application routines and serve as reusable building blocks that the online phase composes via MCTS into executable paths for new instructions.
>
> **W2 (Limited generalization) & W3 (Offline cost).**
> We address generalization from two aspects. First, even when action-level paths cannot be directly reused (simulating scenarios such as software updates or unseen tasks), our KG still provides valuable guidance through task-level operational logic (workflow structure between sub-goals) and action-level element knowledge (functional descriptions of UI elements). To validate this, we held out 25% of tasks from offline exploration, making no pre-explored paths available at inference. As shown in Table R12, the KG-augmented agent still significantly outperforms the baseline, confirming generalization from the structured knowledge rather than mere path replay.
> **Table R12: Performance on held-out tasks (25%, no pre-explored paths).**
>
> | Method            | AndroidWorld | MobileMiniWob++ |
> | ----------------- | ------------ | --------------- |
> | Baseline (w/o KG) | 6.9%         | 29.2%           |
> | EAM (w/ KG)       | 13.8%        | 45.8%           |
>
> Second, even if constructing the KG from scratch for an entirely new application is required, the cost is practically acceptable. We provide a full cost breakdown in Table R13 and R14.
> **Table R13: Average offline exploration cost.**
>
> | Metric         | Per App | Per Task |
> | -------------- | ------- | -------- |
> | Time (s)       | 3768.4  | 724.7    |
> | Token Cost (K) | 1141.4  | 219.5    |
>
> **Table R14: Full cost analysis of EAM on AndroidWorld.**
>
> | Phase              | Metric                  | Cost   |
> | ------------------ | ----------------------- | ------ |
> | DFS Exploration  (offline) | Time per app (s)        | 3768.4 |
> | DFS Exploration (offline) | Token cost per app (K)  | 1141.4 |
> | KG Construction (offline) |  Time per app (s)   | 1019.2 |
> | KG Construction (offline) | Token cost per app (K)  | 214.8 |
> | Online (per step)  | Token cost per step (K) | 53.3  |
>
> The offline cost including exploration and KG contruction is a one-time investment. The 53.3K per-step total in Table R14 includes both online inference (8.3K/step as in our paper) and amortized offline overhead. As more tasks are executed, the amortized offline portion diminishes rapidly and the total converges toward 8.3K, confirming that EAM's cost is dominated by lightweight online inference.
>
> **W4 (Q-model training overhead).**
> We train the Q-model via iterative self-training on 4×A800-80GB GPUs using Qwen2.5-3B-instruct as the base model. The training consists of 4 rounds, with each round containing 5,000–6,000 self-sampled training examples (the exact count depends on MCTS search trajectories on the KG per round). As shown in Fig. 2 of our paper, we analyze the sensitivity to training rounds from two perspectives: (1) task success rate, and (2) the optimal action margin Delta*_min on the critical path (i.e., the Q-value gap between the optimal and second-best action), which directly reflects the model's discriminability for optimal actions. Both metrics improve steadily across rounds and converge by round 4. The training curves in Appendix B (Fig. 6) further show that the loss decreases consistently within each round, with different curves representing different training rounds. The entire self-training process is a one-time cost per application and is modest relative to the offline exploration overhead.

---

> > ### Author Rebuttal · Reviewer_iY2H · 2026-04-03
> >
> > Several concerns remain only partially addressed. The generalization evaluation is limited to held-out tasks within the same environment, rather than across unseen applications. The offline exploration and knowledge graph construction still incur substantial per-environment cost (e.g., ~1 hour per app even on AndroidWorld, which is still a relatively controlled benchmark and likely much simpler than many real-world websites or applications). This makes it unclear how the exploration cost would scale in more realistic environments with larger and more diverse state spaces. The amortization argument may also be less compelling in settings involving many dynamic or previously unseen interfaces. In addition, the reported efficiency mainly reflects online inference, while the full cost including offline stages and training is still less clearly characterized.

---

> > > ### Author Response · Authors · 2026-04-04
> > >
> > > Thank you for the continued engagement.
> > >
> > > **On generalization to unseen applications.**
> > > We would like to respectfully clarify the positioning of our work. EAM's contribution centers on shifting GUI agents from step-wise LLM reasoning to a retrieval-and-execution paradigm — one-shot executable path extraction from a structured KG followed by sequential execution. Unlike RAG-based approaches that may generalize more broadly but suffer from unreliable in-context grounding and cumulative per-step overhead, our approach achieves its efficiency and reliability precisely because the KG faithfully models the target app's dynamics, which makes generalization to entirely unseen applications (without any exploration) outside the intended scope. Nonetheless, the Q-model does learn transferable priors: a Q-model trained solely on AndroidWorld improves performance on MobileMiniWoB++ and DroidTask without app-specific KG (Figure 3 in our paper), so adapting to a new application only requires building the KG while directly reusing the Q-model.
> > >
> > > **On benchmark representativeness.**
> > > AndroidWorld, DroidTask, and MobileMiniWoB++ are standard benchmarks widely adopted in the community — e.g., OSCAR (ICLR '25), GUI-Explorer (ACL '25), EcoAgent (AAAI '26) on AndroidWorld; AutoDroid (MobiCom '24), AutoDroid-V2 (MobiSys '25), UICOMPASS (EMNLP '25) on DroidTask. Industry models prioritizing practical deployment (UI-TARS, GUI-Owl) also adopt AndroidWorld as a key benchmark. AndroidWorld itself runs on a full Android emulator with real-world apps (Clock, Calendar, Expense, Broccoli, OsmAnd, etc.) and dynamically parameterized task variants, providing a meaningful approximation of real-world usage.
> > > To directly address the concern about scalability to more complex environments, we further evaluate EAM on **OSWorld**, a desktop benchmark featuring substantially larger state spaces and more complex interaction patterns. As shown in Table R16, EAM achieves consistent gains on OSWorld.
> > >
> > > **Table R16: Performance on OSWorld.**
> > >
> > > | Method            | Type          | Base Model                         | Success Rate | Tokens/Step (K) |
> > > | ----------------- | ------------- | ---------------------------------- | ------------ | --------------- |
> > > | GPT-5.2           | Cloud         | GPT-5.2                            | 47.3%        | 48.3            |
> > > | Claude-Sonnet-4.5 | Cloud         | Claude-Sonnet-4.5                  | 42.9%        | 47.7            |
> > > | UI-TARS-1.5-7B    | On-device     | UI-TARS-1.5-7B                     | 27.3%        | -               |
> > > | UI-TARS-2B        | On-device     | UI-TARS-2B                         | 3.1%         | -               |
> > > | AppAgentX         | Framework     | GPT-5.2                            | 49.3%        | 12.3            |
> > > | GUI-Explorer      | Framework     | GPT-5.2                            | 52.6%        | 70.2            |
> > > | **EAM (Ours)**    | **Framework** | GPT-5.2, Qwen2.5-3B-ft, UI-TARS-2B | **51.8%**    | **8.9**         |
> > >
> > > We also report the offline cost on OSWorld in Table R17. Due to the significantly more complex state spaces and higher task density per application compared to mobile environments, the offline cost increases accordingly. Nevertheless, the total cost of 23251.67K tokens per app (~ 58.13USD in GPT-4o pricing) remains lower than AutoDroid-V2 and GUI-Explorer on mobile benchmarks.
> > >
> > > **Table R17: Average offline cost breakdown in OSWorld.**
> > >
> > > | Phase             | Metric             | Per App |
> > > | ----------------- | ------------------ | ------- |
> > > | DFS Exploration   | Token Cost (K)     | 18833.29  |
> > > | KG Construction   | Token Cost (K)     | 4418.38    |
> > > | **Total Offline** | **Token Cost (K)** | 23251.67   |
> > >
> > > **On offline cost characterization.**
> > > We would like to respectfully clarify that our rebuttal has provided a detailed breakdown covering the entire offline pipeline: W3 reports the token consumption and wall-clock time for DFS exploration and KG construction per app/task (Table R14), and W4 details the Q-model training setup (4×A800-80GB, Qwen2.5-3B-instruct, 4 rounds, 5,000–6,000 examples/round). We would appreciate it if the reviewer could specify which aspect remains unclear. For broader context, offline exploration cost is an inherent overhead of the explore-exploit paradigm and is rarely quantified in existing works. Among the few that do, AutoDroid-V2 (MobiSys '25) requires ≥82.42USD/app in GPT-4o API calls (excluding initial exploration and SLM training), and GUI-Explorer (ACL '25) yields over 30,000K tokens/app (75.00USD, excluding knowledge base construction) in our reproduction. EAM's task-oriented DFS consumes 1141.4K tokens/app (2.85USD) for the complete pipeline, benefiting from task-goal-guided exploration rather than random traversal. The Q-model's cross-application transferability further reduces adaptation cost (only KG construction needed, no retraining), and a constructed KG is reusable across users and devices sharing the same app version.

---

### Official Review · Reviewer_WKRr · 2026-03-12

**Soundness:** 3
**Presentation:** 3
**Significance:** 3
**Originality:** 3
**Overall Recommendation:** 5
**Confidence:** 5

**Summary:**

The paper introduces Executable Agentic Memory (EAM), a structured GUI execution knowledge graph that shifts mobile GUI agents from step-by-step action generation to a retrieval-and-execution paradigm for more efficient and reliable long-horizon task completion.

The method builds the knowledge graph offline through task-oriented DFS exploration, state/element deduplication, and action-group mining that compresses frequent multi-step routines into reusable high-level actions. At inference time, EAM uses a lightweight Q-model to guide MCTS over the graph, enabling efficient extraction of executable paths while requiring only limited cloud-model involvement for final plan filtering. Experiments on AndroidWorld, MobileMiniWob++, and DroidTask show that EAM substantially improves success rate over baselines such as UI-TARS-7B, while also reducing latency and token cost.

**Compliance With Llm Reviewing Policy:**

Affirmed.

**Final Justification:**

During the rebuttal, authors provided detailed clarifications and exps results, I think most of my concerns are addressed. I would raise my score from 4 to 5

**Key Questions For Authors:**

Overall, I enjoy reading this paper. The following questions could be considered to in further improve this work:

- The success rate on AndroidWorld (52.6%) does not appear highly competitive. Is this due to the use of a 2B language model? If so, it would be helpful if the authors could report the best results using one of the most powerful mobile GUI agents currently available.

- I agree that memory construction and utilization are important for mobile GUI agents. However, relying solely on memory for GUI operations may be misaligned with the ultimate goal of LLM-powered agents, where long-tailed dynamics can be handled at runtime. Therefore, the authors may consider how the proposed methods could be better integrated with ReAct-like GUI agents.

- Some technical details should be clarified, particularly how the correctness and completeness of the traces explored during the offline phase are validated.

- The system overhead is not discussed. As mentioned in AutoDroid, random exploration can be very costly, often requiring tens of hours to explore a single application. How does EAM compare in this regard?

**Limitations:**

yes

**Strengths And Weaknesses:**

Strengths:
- The problem is well motivated and practically important.
- The methods are technically novel. The combination of an executable GUI knowledge graph, task-oriented DFS exploration, action-group mining, and Q-guided MCTS forms a coherent and systematize framework rather than an engineering trick.
- The performance looks good, especially for on-device agents.
- The theoretical analysis is inspiring.

Weakness:
- The obtained improvement might be marginal, i.e., 52.6% success rate on AndroidWorld is not that competitive.
- Some key technical details are still unclear, e.g., how to validate the traces collected in the offline stage and how to guarantee the completeness of the offline KG.
- The methods heavily depend on offline constructed KG, thus the robustness to UI dynamics might be limited.
- The evaluation on system overhead is missing, especially offline stage.

---

> ### Author Rebuttal · Authors · 2026-03-31
>
> We sincerely thank the reviewer for the positive assessment and constructive suggestions.
>
> **Q1 (Performance with stronger models).**
> The 52.6% reported in our paper uses a 2B on-device execution model, prioritizing deployment efficiency. As EAM is a plug-and-play framework, we can seamlessly integrate stronger models. Table R8 reports results on AndroidWorld with more powerful baselines and EAM configurations.
> **Table R8: AndroidWorld results with stronger models.**
>
> | Method                         | Success Rate |
> | ------------------------------ | ------------ |
> | GPT-5.2                        | 56.9%        |
> | Claude-Sonnet-4.5              | 54.3%        |
> | UI-TARS-7B-DPO                 | 30.1%        |
> | GUI-Owl-7B                     | 44.0%        |
> | EAM (GPT-5.2 + UI-TARS-7B-DPO) | 61.2%        |
> | EAM (GPT-5.2 + GUI-Owl-7B)     | 66.4%        |
>
> **Q2 (Integration with ReAct-like agents).**
> We fully agree with the reviewer. In fact, EAM is designed as a pluggable module that integrates into existing ReAct frameworks rather than replacing them. The workflow operates in two modes: (1) When the cloud LLM judges the extracted KG path as directly reusable for the current task, the on-device agent executes the path with parameterized actions, achieving maximum efficiency. (2) When no reusable path is available, the framework falls back to ReAct-style step-wise decision-making, augmented with task-level logic knowledge (workflow structure) and action-level element information from the KG as context. As shown in Table R9, even in this fallback mode, KG-augmented agents significantly outperform the vanilla baseline, confirming that the knowledge provides substantial guidance for runtime reasoning.
> **Table R9: Fallback mode performance (25% held-out tasks, no reusable paths).**
>
> | Method                            | AndroidWorld | MobileMiniWob++ |
> | --------------------------------- | ------------ | --------------- |
> | Baseline (w/o KG)                 | 6.9%         | 29.2%           |
> | EAM fallback (KG-augmented ReAct) | 13.8%        | 45.8%           |
>
> **Q3 (Correctness and completeness of offline traces).**
> Our DFS exploration is task-oriented with a built-in state validation mechanism. After each sub-goal is executed, a state checker evaluates the current state and determines one of three outcomes: (1) COMPLETE — the task goal is achieved; (2) CONTINUE — the sub-goal succeeded but further exploration is needed; (3) BACKTRACK — the current state deviates from the target, triggering a rollback to the previous depth to explore alternative paths. This systematic mechanism ensures that the exploration covers, with high probability, all states necessary for task completion along different execution branches. To quantify the relationship between exploration coverage and task performance, we vary the proportion of tasks explored per app. As shown in Table R10, success rate scales positively with coverage, and even 75% coverage yields substantial gains.
> **Table R10: Success rate vs. offline exploration ratio.**
>
> | Exploration Ratio | AndroidWorld | MobileMiniWob++ |
> | ----------------- | ------------ | --------------- |
> | 0% (no KG)        | 6.9%         | 31.5%           |
> | 75%               | 42.2%        | 66.7%           |
> | 100% (full EAM)   | 52.6%        | 76.1%           |
>
> **Q4 (System overhead).**
> Unlike AutoDroid's random exploration which can require tens of hours per app, our task-oriented DFS is significantly more efficient due to sub-goal-guided traversal and state deduplication. We report the offline overhead on AndroidWorld in Table R11.
> **Table R11: Average offline cost on AndroidWorld.**
>
> | Phase              | Metric                  | Cost   |
> | ------------------ | ----------------------- | ------ |
> | DFS Exploration  (offline) | Time per app (s)        | 3768.4 |
> | DFS Exploration (offline) | Token cost per app (K)  | 1141.4 |
> | KG Construction (offline) |  Time per app (s)   | 1019.2 |
> | KG Construction (offline) | Token cost per app (K)  | 214.8 |
>
> The average offline time including exploration and KG construction is ~ 80 minutes per app, substantially lower than random exploration approaches. This is a one-time cost: as the number of online tasks grows, the amortized offline overhead per task diminishes rapidly and becomes negligible relative to the online inference cost reported in the paper (8.3K tokens/step, 2.8s latency).

---

> > ### Author Rebuttal · Reviewer_WKRr · 2026-04-01
> >
> > Thanks for the efforts and reply. I believe these results and clarifications have addressed most of my concerns.
> >
> > One minor question left might be: how the state checker works, say, how does it determine whether to mark a step as complete, continue, or backtrack, and how is the correctness of these decisions ensured? In addition, how does the system roll back to the exact previous state on Android when backtracking is required?

---

> > > ### Author Response · Authors · 2026-04-02
> > >
> > > Thank you for the follow-up question.
> > >
> > > **On the state checker:** Our state checker is inspired by how humans interact with unfamiliar interfaces: identify a promising element, interact, assess whether the outcome aligns with expectations, and backtrack to try alternatives if not. Our design formalizes this cognitive loop — at each DFS depth, the cloud LLM receives the current UI state, the executed sub-goal, and the task goal, then judges whether to CONTINUE, BACKTRACK, or mark COMPLETE (prompt details in Appendix. D).
> > > Since intermediate exploration steps lack ground-truth labels, directly quantifying the state checker's accuracy is challenging. Instead, we conduct an ablation where only the state checker model is varied, while the sub-goal generation and task execution model are both fixed as GPT-5.2. As shown in Table R15, the state checker is critical: a weaker checker misleads the explorer in two failure modes: (1) failing to BACKTRACK when the sub-goal has already deviated from the task goal, causing continued exploration into unproductive branches (wasting tokens while introducing noise into the KG), and (2) prematurely terminating correct sub-goal exploration, directly ignoring potentially valid paths. Both failure modes degrade KG quality and consequently online inference performance.
> > >
> > > **Table R15: Impact of state checker model (AndroidWorld, sub-goal generation & execution fixed as GPT-5.2).**
> > >
> > > | State Checker Model     | Success Rate |
> > > | ----------------------- | ------------ |
> > > | Qwen2.5-VL-72B-instruct | 49.1%        |
> > > | GPT-4o                  | 55.2%        |
> > > | GPT-5.2                 | 59.5%        |
> > >
> > > **On rollback:** Following GUI-Explorer's state recovery approach, our DFS explorer stores the complete action history from the initial state to the current depth. When BACKTRACK is triggered, the environment is reset to the initial state and the stored actions are sequentially replayed up to the parent depth. Although replay introduces additional action executions, these are purely sequential environment interactions that incur no extra model token consumption, and the accuracy of state recovery is fully guaranteed.

---

### Official Review · Reviewer_U7Qd · 2026-03-14

**Soundness:** 3
**Presentation:** 3
**Significance:** 2
**Originality:** 3
**Overall Recommendation:** 4
**Confidence:** 3

**Summary:**

This paper presents EAM, a framework designed to improve both task completion efficiency and generalization capability of mobile GUI agents. The approach is organized around two phases. In the offline phase, the system autonomously explores target applications via a DFS strategy, constructing a knowledge graph in which UI states serve as nodes and action transitions serve as edges. High-frequency action sequences are then compressed using a BPE-inspired mechanism to form higher-level composite actions. In the online inference phase, the system performs path planning over the knowledge graph using MCTS in combination with a lightweight Q-value model, enabling task completion with only a minimal number of cloud API calls. The paper further introduces a self-training mechanism based on Bellman backup, which provides fine-grained soft-label supervision for intermediate steps, and offers theoretical support in the form of a bias consistency guarantee for the Q-model and a sample complexity bound for path recovery. Experiments conducted on 3 benchmarks — AndroidWorld, DroidTask, and MobileMiniWob++ — demonstrate that EAM achieves favorable improvements in both success rate and efficiency compared to existing methods.

**Compliance With Llm Reviewing Policy:**

Affirmed.

**Final Justification:**

Thanks for the clarification, which addressed most of my concerns. I'd accordingly raise my score.

**Key Questions For Authors:**

- This paper reports a ~6× reduction in token consumption at inference time, but the offline phase also involves substantial GPT-4o calls for DFS exploration and sub-goal generation. Could the authors provide a full cost analysis that includes the offline phase and revisit the efficiency comparison on a total-cost basis?

- Since the knowledge graph is built by exploring task-relevant paths offline, inference-time success may partly reflect retrieval over pre-explored routes rather than genuine generalization. Could the authors evaluate the system on tasks or apps entirely excluded from the offline exploration phase, to more rigorously validate the generalization claims?

- The completeness of the knowledge graph depends heavily on the coverage achieved during DFS exploration. Could the authors provide a quantitative coverage analysis and examine whether gaps in coverage correlate with task failures? This would offer important insight into the robustness of the approach.

- The paper acknowledges that the knowledge graph becomes outdated after app updates, but does not discuss any mitigation strategies. Could the authors elaborate on the cost of rebuilding the graph after a typical update, and whether any incremental update mechanism has been considered?

- Theorems 5.2 and 5.3 build on a Q-value formulation borrowed from prior work, and the key conclusions —more data improves accuracy and reduces required MCTS simulations — appear to follow from fairly standard statistical learning arguments. Could the authors clarify what is technically novel in these results, and whether they yield any non-trivial insights specific to the GUI agent setting?

**Limitations:**

yes

**Strengths And Weaknesses:**

**Strengths:**

- Modeling GUI operations as a state machine is a well-motivated design choice. Since app screen transitions naturally follow the structure of a deterministic finite automaton, representing them as a knowledge graph appears to be a more principled approach than having an LLM guess at each step.

- The reported 6× reduction in token consumption and a latency of 2.8s suggest that the framework could be competitive in real-world deployment scenarios.

- Adapting the Byte Pair Encoding idea to GUI action sequence compression is a creative and thoughtful contribution. Leveraging statistical frequency rather than relying on LLM summarization to discover high-level actions is both elegant and more robust.

- The paper provides bias consistency guarantees for the Q-model and a sample complexity bound for path recovery, lending the work a reasonable degree of theoretical grounding.

- The figures throughout the paper are well-designed and visually clear.

**Weaknesses:**

- The offline nature of the knowledge graph construction raises practical concerns. Since the graph becomes stale upon any app update, and the paper acknowledges this limitation without offering a mitigation strategy, the real-world applicability of the approach remains unclear. More importantly, given that task-relevant paths are explored offline in advance, the high success rate at inference time may partly reflect the system navigating a pre-explored space rather than genuinely generalizing, and the paper does not sufficiently evaluate performance on entirely novel tasks.

- While the paper emphasizes the token efficiency gains at inference time, the substantial GPT-4o usage incurred during the offline graph construction phase is not accounted for in the cost analysis. This omission makes the efficiency comparison somewhat misleading.

- The mathematical derivations surrounding Theorem 5.2 and 5.3 occupy considerable space, yet the core conclusions — that more training data leads to better model accuracy and fewer MCTS simulations — are fairly intuitive results. Furthermore, the Q-value formulation relies on a random policy definition borrowed from prior work, which somewhat limits the theoretical novelty of these contributions.

- The quality of the knowledge graph is inherently tied to the quality of sub-goals generated by GPT-4o during DFS exploration. If certain critical paths are missed during this phase, the graph will contain blind spots. Unfortunately, the paper does not analyze coverage rates or quantify how much of the task space may have been overlooked.

- The central idea of building an offline knowledge base and retrieving from it at inference time is not entirely new, as similar paradigms have been explored in prior work. The main contributions appear to lie in the combination of MCTS with a knowledge graph and the BPE-inspired action compression, which, while useful, gives the overall work more of an engineering flavor than a fundamentally novel theoretical one.

---

> ### Author Rebuttal · Authors · 2026-03-31
>
> We thank the reviewer for the insightful comments.
>
> **Q1 (Full cost analysis including offline phase).**
> We provide a full cost breakdown in Table R5.
> **Table R5: Full cost analysis of EAM on AndroidWorld.**
>
> | Phase              | Metric                  | Cost   |
> | ------------------ | ----------------------- | ------ |
> | DFS Exploration  (offline) | Time per app (s)        | 3768.4 |
> | DFS Exploration (offline) | Token cost per app (K)  | 1141.4 |
> | KG Construction (offline) |  Time per app (s)   | 1019.2 |
> | KG Construction (offline) | Token cost per app (K)  | 214.8 |
> | Online (per step)  | Token cost per step (K) | 53.3  |
>
> The offline cost including exploration and KG construction is a one-time investment. The 53.3K per-step total includes both online inference (8.3K/step as in our paper) and amortized offline overhead. As tasks grow, the amortized offline portion diminishes and the total converges toward 8.3K, confirming that EAM's cost is dominated by lightweight online inference.
>
> **Q2 (Generalization beyond pre-explored tasks).**
> We held out 25% of tasks from offline exploration, ensuring no pre-explored action paths were available at inference time. As shown in Table R6, EAM with KG still substantially outperforms the baseline on both benchmarks, because the KG encodes task-level semantic knowledge (high-level workflow structure) that remains transferable to unseen tasks sharing similar operational logic.
> **Table R6: Performance on held-out tasks (25%, no pre-explored paths).**
>
> | Method            | AndroidWorld | MobileMiniWob++ |
> | ----------------- | ------------ | --------------- |
> | Baseline (w/o KG) | 6.9%         | 29.2%           |
> | EAM (w/ KG)       | 13.8%        | 45.8%           |
>
> **Q3 (Coverage analysis and correlation with task failures).**
> Since the benchmarks only provide binary task-completion rewards without intermediate step-level annotations, directly quantifying fine-grained path coverage rates is difficult. However, we can analyze the relationship between exploration coverage and task success by controlling the proportion of tasks used for offline exploration per app. Specifically, for each app we randomly select a certain percentage of its associated tasks to perform DFS exploration and construct the KG, then evaluate on all tasks. As shown in Table R7, we vary this ratio from 0% (no exploration, equivalent to baseline) to 100% (full EAM). The results demonstrate a clear positive correlation between exploration coverage and success rate, and that even 75% coverage yields substantial gains over the baseline.
> **Table R7: Success rate vs. offline exploration ratio.**
>
> | Exploration Ratio | AndroidWorld | MobileMiniWob++ |
> | ----------------- | ------------ | --------------- |
> | 0% (no KG)        | 6.9%         | 31.5%           |
> | 75%               | 42.2%        | 66.7%           |
> | 100% (full EAM)   | 52.6%        | 76.1%           |
>
> **Q4 (Adaptability to app updates and incremental updates).**
> App updates typically preserve operation logic while partially altering layouts, so task-level KG knowledge remains effective (Table R6). If full re-exploration is needed, the cost is acceptable (Table R5). Moreover, since our KG is a state machine with explicit nodes and edges, it naturally supports incremental updates: new states discovered online can be inserted and obsolete nodes pruned without full rebuilding. We plan to develop a systematic online KG update mechanism in future work.
>
> **Q5 (On novelty and theoretical contribution, also addressing W3 & W5).**
> Unlike prior works that inject retrieved knowledge as per-step context (RAG paradigm), EAM treats the KG as a state machine for one-shot path extraction followed by sequential execution, fundamentally improving efficiency and reliability. The key challenge is guaranteeing accurate path extraction. Our design is coherent: BPE-based action compression yields a compact search space that reduces MCTS complexity, while Q-guided MCTS leverages the KG's structural properties for provably optimal extraction.
> To rigorously establish the feasibility of this design beyond intuition, we provide theoretical guarantees. A key challenge is that unlike standard MCTS where node values come from unbiased Monte Carlo simulations, our design uses a learned Q-model as the value estimator, which inevitably introduces bias. We address this by jointly analyzing the coupling between the learned random-policy Q-value and MCTS: Theorem 5.2 establishes when the Q-model preserves correct action rankings on the critical set (epsilon_bias < Delta*_min/2), and Theorem 5.3 derives the MCTS sample complexity as a function of both Q-model bias and KG structure (branching factor K, horizon H). This coupled analysis is absent in standard MCTS theory and statistical learning. These results directly guided our self-training data volume and MCTS iteration count in experiments.

---

> > ### Author Rebuttal · Reviewer_U7Qd · 2026-04-03
> >
> > Thanks for the efforts and the detailed reply with additional experimental results. The clarifications on offline cost, held-out-task evaluation, and the discussion of the theoretical contribution are helpful, and they improve the paper. However, I still have some reservations about the core scope and practical generalization of this method. In particular, the additional generalization evidence is still mainly based on held-out tasks within largely the same environments, rather than truly unseen applications/interfaces. I also remain somewhat concerned that the framework’s effectiveness depends substantially on environment-specific offline exploration and knowledge graph construction, which may limit scalability in more dynamic real-world settings. While the cost analysis is useful, the overall efficiency claim still appears to rely heavily on amortizing a nontrivial offline cost, especially for scenarios with fewer tasks.
> > .

---

> > > ### Author Response · Authors · 2026-04-04
> > >
> > > Thank you for the continued engagement.
> > >
> > > **On generalization to unseen applications.**
> > > We appreciate this concern, but would like to clarify the scope of our work. EAM is designed for efficient and reliable *executable memory reuse within explored applications*. As discussed in our paper, unlike RAG-based contextual knowledge augmentation, where retrieved semantic knowledge may generalize across applications but relies on the model's in-context understanding at every step (which is unreliable) and incurs a full retrieval-generation pipeline per action (which is inefficient) — EAM's core contribution lies in efficient and reliable one-shot executable path extraction followed by sequential execution, both of which are grounded in the KG's modeling of the current app's dynamics. This design choice inherently means that direct transfer to unseen applications without any exploration is not the primary focus of this work.
> > >
> > > That said, while the KG itself is app-specific, the Q-model trained on it does exhibit generalization to unseen applications. As shown in Figure 3 of our paper, a Q-model trained solely on AndroidWorld yields clear performance gains on both MobileMiniWob++ and DroidTask. This suggests that the Q-model learns generalizable structural priors (e.g., action ordering preferences, sub-goal decomposition patterns) that transfer across different applications, even without app-specific KG construction. Consequently, adapting EAM to a new application only requires constructing the app-specific KG, without retraining the Q-model from scratch, which significantly reduces the adaptation cost.
> > >
> > > **On offline cost.**
> > > Offline exploration cost is a shared overhead inherent to the explore-exploit paradigm and is rarely reported in existing works. We provide a full cost breakdown in Table R5 for transparency. To contextualize our cost: AutoDroid-V2 (MobiSys '25), as one of the few works that discloses offline costs, requires ≥82.42USD/app in GPT-4o API calls for its offline documentation, data synthesis, and validation stages — notably excluding the initial random exploration cost and subsequent SLM training cost. GUI-Explorer (ACL'25), one of our baselines, does not report offline exploration costs in the original paper; our reproduction yields over 30,000K tokens per app (~ 75.00USD in GPT-4o pricing) in AndroidWorld, again excluding the cost of knowledge base construction. In comparison, EAM's task-oriented DFS exploration consumes 1141.4K tokens per app (2.85USD), covering the full KG construction pipeline. This efficiency benefits from our task-oriented DFS design, which leverages explicit task goals to guide exploration rather than random or exhaustive traversal. Furthermore, as discussed above, the Q-model's cross-environment generalization means that adapting to a new environment only requires KG construction without model retraining, further reducing the practical adaptation cost. Moreover, once constructed, a KG can be reused across different users and devices running the same app version, since the KG encodes app-level dynamics with relative UI coordinates rather than device-specific absolute positions. This means the one-time offline cost is amortized not only across tasks but also across the entire user base sharing the same app environment.

---

### Official Review · Reviewer_1w1g · 2026-03-18

**Soundness:** 3
**Presentation:** 3
**Significance:** 3
**Originality:** 3
**Overall Recommendation:** 4
**Confidence:** 4

**Summary:**

The manuscript proposes Executable Agentic Memory, a framework designed to transition GUI agents from a step-wise, model-centric generation paradigm to a retrieval-and-execution process. To address the compounding errors and high costs of long-horizon tasks, the framework introduces an offline phase that constructs a GUI Logic Knowledge Graph via state-aware DFS exploration.  This offline phase also employs a statistical, BPE-inspired action group mining technique to compress multi-step routines into reusable high-level actions. During online inference, the framework utilizes a lightweight Q-model to guide an MCTS over the Knowledge Graph, allowing the agent to extract executable, high-reward paths rather than blindly generating actions at every step. The authors also provide theoretical guarantees, establishing bias-consistency for the Q-model and sample complexity bounds for MCTS path recovery.

**Compliance With Llm Reviewing Policy:**

Affirmed.

**Final Justification:**

Thanks for the authors' rebuttal and supplemental experiments. They addressed most of my concerns. I'd accordingly raise my score.

**Key Questions For Authors:**

Please see weaknesses.

**Limitations:**

Yes.

**Strengths And Weaknesses:**

Strengths
- Strong Theoretical Foundation: Unlike many contemporary empirical agent papers, this work rigorously grounds its methodology in theory. The inclusion of a bias-consistency guarantee for the learned Q-model and sample complexity bounds for MCTS provides a solid mathematical justification for the framework's efficiency.
- High Operational Efficiency: The framework is highly practical for real-time deployment. By relying on a lightweight Q-model for scoring and restricting heavy LLM usage to a single path-filtering call, the system achieves an impressive average latency of 2.8 seconds and cuts token costs by 6x relative to cloud-based alternatives.

Weaknesses
- Offline Exploration Scalability: The state-aware DFS exploration strategy might struggle with infinite-scroll pages or dynamically loaded content, where the state space is theoretically unbounded. Building a comprehensive Knowledge Graph in such environments could lead to an explosion in offline computational costs. Also, how such paradigm can swiftly be applied to updated softwares or whole-new softwares are not discussed or framed.
- Limited Baseline Scaling: The paper does not push the baselines extensively. The authors are using small (2B, 3B, 7B) or relatively older models (such as GPT-4o) as baselines, and they primarily prove the effectiveness of the framework with GPT-4o and Qwen-3B. However, it remains unknown how the framework performs or compares when integrated with larger models. It is especially important to demonstrate that the framework can be scaled up on larger models with stronger foundation performances.
- Omission of Key Benchmarks: Another popular GUI agent benchmark, OSWorld, is not explored. Failing to evaluate on this highly rigorous, multi-modal desktop environment leaves a critical gap in assessing the cross-platform robustness and general applicability of the framework.

---

> ### Author Rebuttal · Authors · 2026-03-31
>
> We sincerely thank the reviewer for the constructive feedback. We address each concern below.
>
> **W1 (Offline Exploration Scalability).**
> **On state space explosion:** We address this from two aspects. (1) Our offline exploration is task-oriented rather than exhaustive. While an application's state space is theoretically unbounded, real-world users navigate a limited set of repetitive functional paths. Our DFS is guided by task-relevant sub-goals, constraining exploration to the practically reachable portion. These trajectories capture core application routines and serve as reusable building blocks. (2) Within this already-constrained space, our KG construction further applies semantic-level state/element deduplication, merging functionally equivalent states and redundant UI elements. This substantially compresses the final KG relative to raw trajectories.
>
> **On adaptability to software updates:** App updates may alter page layouts, but typically preserve the underlying operation logic. Our KG encodes both task-level semantic logic and action-level element information. While the latter may become stale after updates, the former remains robust. To validate this, we held out 25% of tasks from offline exploration (no pre-explored action paths available) and evaluated on AndroidWorld and MobileMiniWob++. As shown in Table R1, EAM with KG still significantly outperforms the baseline, confirming that task-level knowledge provides substantial guidance even without directly reusable action paths.
> **Table R1: Performance on held-out tasks (25%, excluded from offline exploration).**
>
> | Method                             | AndroidWorld | MobileMiniWob++ |
> | ---------------------------------- | ------------ | --------------- |
> | Baseline (w/o KG)                  | 6.9%         | 29.2%           |
> | EAM (w/ KG, no pre-explored paths) | 13.8%        | 45.8%           |
>
> **On entirely new software:** We report offline exploration cost in Table R2. The cost per app is a one-time investment amortized across all subsequent task executions.
> **Table R2: Average offline cost on AndroidWorld.**
>
> | Phase             | Metric             | Per App | Per Task |
> | ----------------- | ------------------ | ------- | -------- |
> | DFS Exploration   | Time (s)           | 3768.4  | 724.7    |
> | DFS Exploration   | Token Cost (K)     | 1141.4  | 219.5    |
> | KG Construction   | Time (s)           | 1019.2   | 196     |
> | KG Construction   | Token Cost (K)     | 214.8   | 41.3     |
>
>
>
> **W2 (Limited Baseline Scaling).**
> We clarify that models in EAM serve distinct roles: (1) a cloud LLM (GPT-4o) for offline exploration and a single online path-selection call; (2) a fine-tuned Qwen2.5-3B-instruct model as a lightweight Q-model scorer for MCTS; (3) an on-device UI-TARS-2B for task execution. Small models for (2) and (3) are a deliberate design choice for on-device deployment, not a limitation. By offloading knowledge into a structured KG, lightweight on-device models achieve both significant performance gains and high efficiency (2.8s latency, ~6× token reduction) with only one cloud API call. Furthermore, EAM is model-agnostic and plug-and-play. As shown in Table R3, replacing GPT-4o with stronger models yields further improvements, confirming the framework's scaling capability.
> **Table R3: EAM with different cloud models.**
>
> | Cloud Model       | AndroidWorld | MobileMiniWob++ |
> | ----------------- | ------------ | --------------- |
> | GPT-4o            | 52.6%        | 76.1%           |
> | GPT-5.2           | 59.5%        | 81.3%           |
> | Claude-Sonnet-4.5 | 57.8%        | 80.2%           |
>
> **W3 (Omission of OSWorld).**
> We have conducted additional experiments on OSWorld. As shown in Table R4, EAM achieves competitive performance (51.8%) comparable to GUI-Explorer (52.8%) while consuming significantly fewer tokens per step (8.9K vs. 70.2K), demonstrating effective cross-platform generalization with efficiency advantages preserved.
> **Table R4: Performance on OSWorld.**
>
> | Method            | Type          | Base Model                         | Success Rate | Tokens/Step (K) |
> | ----------------- | ------------- | ---------------------------------- | ------------ | --------------- |
> | GPT-5.2           | Cloud         | GPT-5.2                            | 47.3%        | 48.3            |
> | Claude-Sonnet-4.5 | Cloud         | Claude-Sonnet-4.5                  | 42.9%        | 47.7            |
> | UI-TARS-1.5-7B    | On-device     | UI-TARS-1.5-7B                     | 27.3%        | -               |
> | UI-TARS-2B        | On-device     | UI-TARS-2B                         | 3.1%         | -               |
> | AppAgentX         | Framework     | GPT-5.2                            | 49.3%        | 12.3            |
> | GUI-Explorer      | Framework     | GPT-5.2                            | 52.6%        | 70.2            |
> | **EAM (Ours)**    | **Framework** | GPT-5.2, Qwen2.5-3B-ft, UI-TARS-2B | **51.8%**    | **8.9**         |

---

> > ### Author Rebuttal · Reviewer_1w1g · 2026-04-02
> >
> > Thanks for the authors' rebuttal and supplemental experiments. They addressed most of my concerns. I'd accordingly raise my score.

---

### Decision · Program_Chairs · 2026-04-30

**Decision:**

Accept (regular)

**Comment:**

This paper addresses an important problem in GUI agents and was viewed by the reviewers as technically solid, well motivated, and practically meaningful. The proposed executable memory framework was seen as a coherent contribution, with strong efficiency benefits and useful theoretical support.

The main concerns were about offline exploration cost and generalization, but the rebuttal substantially addressed these points through additional experiments, stronger comparisons, and clearer cost analysis. Reviewers generally found the response convincing, and several raised their scores accordingly.

Overall, the reviewer consensus is positive, and I recommend acceptance.